# *Selenomonas sputigena* acts as a pathobiont mediating spatial structure and biofilm virulence in early childhood caries

Hunyong Cho[1,13], Zhi Ren [2,13], Kimon Divaris [3,4] ✉, Jeffrey Roach[5,6], Bridget M. Lin[1], Chuwen Liu[1], M. Andrea Azcarate-Peril[6,7], Miguel A. Simancas-Pallares [3], Poojan Shrestha [3,4], Alena Orlenko[8], Jeannie Ginnis[3], Kari E. North [4], Andrea G. Ferreira Zandona[9], Apoena Aguiar Ribeiro[10], Di Wu [1,11] ✉ & Hyun Koo [2,12] ✉

*Streptococcus mutans* has been implicated as the primary pathogen in childhood caries (tooth decay). While the role of polymicrobial communities is appreciated, it remains unclear whether other microorganisms are active contributors or interact with pathogens. Here, we integrate multi-omics of supragingival biofilm (dental plaque) from 416 preschool-age children (208 males and 208 females) in a discovery-validation pipeline to identify disease-relevant inter-species interactions. Sixteen taxa associate with childhood caries in metagenomics-metatranscriptomics analyses. Using multiscale/computational imaging and virulence assays, we examine biofilm formation dynamics, spatial arrangement, and metabolic activity of *Selenomonas sputigena, Prevotella salivae* and *Leptotrichia wadei*, either individually or with *S. mutans*. We show that *S. sputigena*, a flagellated anaerobe with previously unknown role in supragingival biofilm, becomes trapped in streptococcal exoglucans, loses motility but actively proliferates to build a honeycomb-like multicellular-superstructure encapsulating *S. mutans*, enhancing acidogenesis. Rodent model experiments reveal an unrecognized ability of *S. sputigena* to colonize supragingival tooth surfaces. While incapable of causing caries on its own, when co-infected with *S. mutans, S. sputigena* causes extensive tooth enamel lesions and exacerbates disease severity in vivo. In summary, we discover a pathobiont cooperating with a known pathogen to build a unique spatial structure and heighten biofilm virulence in a prevalent human disease.

Biofilm-forming pathogens have been implicated in a myriad of human infectious diseases and contamination of biomedical devices and implants[1–3]. Although established pathogens display emergent properties that facilitate biofilm lifestyle and virulence expression, they often reside in polymicrobial communities with increasing evidence of inter-species interactions[4,5]. Among various microbiomes, the human plaque biofilm communities harbor diverse microbiota at the oral mucosal barrier and mineralized dental surfaces that play key roles in modulating health and disease yet remain underappreciated[6]. Although microbiome-wide association studies have revealed novel microbial species implicated in oral diseases, most evidence has been generated by small samples of participants recruited from clinics or undergoing dental treatment, whereas the identified species' causal roles remain largely undefined. Comprehensive investigations of the taxonomic and

functional aspects of microbial interactions in oral biofilms among community-based samples are warranted. Such studies may not only reveal additional pathogens with population relevance but can also shed light on novel pathogenetic mechanisms and interspecies interactions.

Dental caries (tooth decay) is a widespread, biofilm-mediated, and diet-modulated disease that affects approximately 600 million children worldwide and remains a major unresolved public health problem[7]. The disease has a complex etiology, but it is generally understood as a host-diet-dependent process of dental tissue demineralization that relies on dysbiotic community and polymicrobial acidogenesis[4,8]. *Streptococcus mutans* is a gram-positive, biofilm-forming, acidogenic and aciduric bacterium that has been clinically associated with childhood caries[9], has been shown to cause caries in animal models[10], and has been established as a keystone oral pathogen[11]. Other bacterial taxa, e.g., *Scardovia wiggsiae* and mitis group streptococci such as *Streptococcus gordonii* and *Streptococcus oralis*, have been reported as collaborating with or antagonizing *S. mutans*[12,13]. Although other microbial species on tooth surfaces have been associated with dental caries[14-17], it remains unclear whether they are active contributors, inactive cohabitants, or they interact with *S. mutans* as pathobionts[18] to promote disease development.

Most evidence to date on the microbial basis of dental caries has been generated from targeted, culture-based, or 16S ribosomal gene-based methods. While providing foundational information, this knowledge base can be substantially augmented using DNAseq-based metagenomics (MTG), as well as RNAseq-based metatranscriptomics (MTX), to gain deeper insights into microbial taxonomy and functional activity. The few studies employing next-generation sequencing have been done in small clinical samples[19,20], and with little or no mechanistic validation. Meanwhile, advances in imaging technology have revealed spatially structured communities in multispecies biofilms, warranting better understanding of the spatial arrangement and positioning (i.e., the biogeography) of the interacting microbes[21,22]. To overcome these limitations and gain fundamental knowledge, we have developed a multi-modality, discovery-validation pipeline integrating clinical data from community-based studies and informatics discovery with in vitro and in vivo experimental models to study disease-associated microbial taxa and their interactions (Fig. 1).

Our data highlight *Selenomonas sputigena*, *Prevotella salivae* and *Leptotrichia wadei* as species with previously unrecognized roles in supragingival biofilms, with functional repertoires and interactions relevant to the pathogenesis of a prevalent childhood dysbiotic disease. We unexpectedly find that the flagellated, gram-negative anaerobic bacterium *S. sputigena*, found in saliva and commonly reported as member of the periodontal niche microbiome[23], has a significant role in the supragingival biofilm microbiome and is strongly associated with childhood dental caries. Here, we show that the motile *S. sputigena* becomes trapped into extracellular glucan-matrix produced by *S. mutans*, loses its motility, but rapidly proliferates to form a multicellular honeycomb-like scaffold that wraps the streptococcal cell clusters and results in enhanced acid production—a key virulence factor in dental caries. We verify experimentally that, although *S. sputigena* alone does not cause caries, it significantly increases disease severity when co-infected with *S. mutans* causing extensive cavitation of tooth enamel in a rodent model. Taken together, our findings reveal a novel pathobiont capable of unique spatial structuring and interspecies cooperation that enhances biofilm virulence—a phenomenon that substantially improves our understanding of childhood dental disease pathogenesis and may have implications for established pathogen and pathobiont interactions in other polymicrobial infections.

## Results

### Established and novel taxa are associated with childhood caries
We carried out comprehensive taxonomic association analyses in MTG (i.e., whole genome sequencing shotgun) and MTX (i.e.,

RNAseq) data and identified 16 bacterial species significantly associated with clinically measured childhood caries experience. We used strict, multiple testing (i.e., false discovery rate, FDR)-controlled, across-trait criteria to identify this set of 16 disease-associated species. Specifically, we required taxa to demonstrate statistically significant associations with quantitative disease experience measured both locally (i.e., number of tooth surfaces with caries experience considering tooth surfaces where plaque was harvested from) and at the person-level (i.e., number of tooth surfaces with caries experience considering the entire dentition), in DNA and RNA data, in the discovery sample set ($n = 300$ children), and with evidence of replication in an independent sample of similarly-aged children ($n = 116$, Fig. 2 and Supplementary Table 1). *S. mutans* (beta = 0.51 ± 0.21, $p = 2.6 \times 10^{-6}$) emerged strongly associated with caries experience as the known and expected suspect. Other significantly associated taxa included *P. salivae* (beta = 0.31 ± 0.14, $p = 8.5 \times 10^{-6}$ in the association of DNA-based species taxonomic abundance with localized caries experience), *S. sputigena* (beta = 0.53 ± 0.22, $p = 3.1 \times 10^{-6}$), *L. wadei* (beta = 0.41 ± 0.15, $p = 2.2 \times 10^{-7}$), *Veillonella atypica* (beta = 0.41 ± 0.18, $p = 1.1 \times 10^{-5}$), *Lachnospiraceae bacterium oral taxon 082* (beta = 0.30 ± 0.15, $p = 1.1 \times 10^{-4}$), *Stomatobaculum longum* (beta = 0.36 ± 0.22, $p = 1.6 \times 10^{-3}$), *Lachnoanaerobaculum saburreum* (beta = 0.43 ± 0.17, $p = 9.6 \times 10^{-7}$), and *Centipeda periodontii* (beta = 0.31 ± 0.21, $p = 3.8 \times 10^{-3}$). These associations were consistent in MTG and MTX data (Supplementary Fig. 1), and, as expected, there was a high degree of correlation between the abundance of taxonomically neighboring significant taxa in MTG and MTX (e.g., *Leptotrichia* spp. and *Prevotella* spp.) (Fig. 3). *S. mutans*, being the only *Streptococcus* among the 16 significant species, showed the smallest correlations with the other 15 taxa. For example, *S. mutans'* mean correlations in MTX were 0.09 both in health and disease, whereas *L. wadei's* were 0.50 and 0.49, and *S. sputigena's* were 0.45 and 0.38, respectively.

### Glycolysis is upregulated in biofilm transcriptomes where novel taxa are implicated
Six pathways emerged as significantly associated with caries experience in MTX: glycolysis IV, pyruvate fermentation to acetate and lactate II, chorismate biosynthesis from 3-dehydroquinate, UDP-N-acetyl-D-glucosamine biosynthesis I, inosine-5′-phosphate biosynthesis III, and superpathway of menaquinol-8 biosynthesis II (Supplementary Table 2). Of note, the first 4 of these pathways involved one or more of the 16 significant species. Glycolysis IV, one of the most abundant pathways in expression abundance (ranked #3 overall) and an important pathway for acidogenesis implicated in disease, involves *S. sputigena*, *L. wadei* and 4 more significant species. Pyruvate fermentation to acetate and lactate II involves *L. saburreum* and its significant down-regulation is consistent with a decrease in the availability of pyruvate in an acidogenic caries-promoting biofilm, wherein it may be preferentially consumed by *S. mutans*. Additionally, the upregulated UDP-N-acetyl-D-glucosamine biosynthesis I, involving *L. wadei*, *C. periodontii*, *S. sputigena*, and *V. atypica*, is important for peptidoglycan synthesis and cell growth. In addition, lactose and galactose degradation I, the pathway with the highest proportional representation of significant species (i.e., *L. wadei* and *S. mutans*) was also upregulated in disease although not statistically significant (Supplementary Fig. 2).

### Three new candidate pathogens nominated for virulence and biofilm studies
Next, we sought to prioritize a subset of species that was practically feasible to carry forward into the multi-modal validation pipeline (Fig. 1 steps 6–8). To select this shortlist of candidates that could be fully characterized in virulence, biofilm, and in vivo experiments in this study, we considered statistical evidence of association in the discovery sample (i.e., *p*-value), evidence independent replication, representation of different genera (i.e., one species per genus), and

① **Clinical and Biospecimen Data Collection**

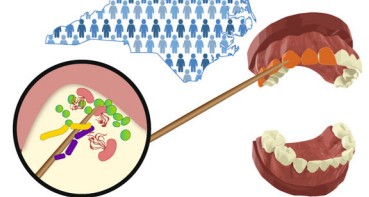

- Community-based sample (N=300)
- 3-5-year-olds (50% with disease)

② **Nucleic Acid Isolation and Sequencing**

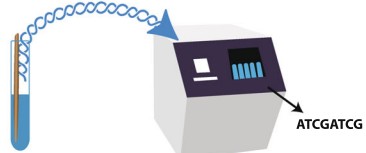

ATCGATCG

- NA Isolation & Library creation
- Illumina sequencing (DNA/RNA)

③ **QA/QC, Alignment, and Annotation**

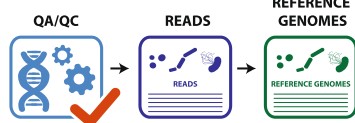

- Demultiplexing, QA/QC
- Taxonomic classification

④ **Informatics and Statistical Inference**

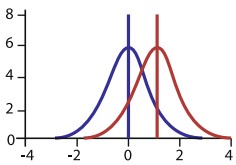

- Normalization/rescaling to TPM
- Abundance and prevalence filtering
- Log-normal linear regression modeling

⑤ **Taxonomic Discovery and Replication**

Discovery n=300 ⟶ Replication n=116

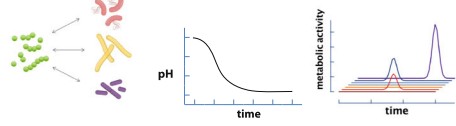

- Localized & person-level traits, DNA/RNA
- FDR-differentially abundant species
- Discovery & replication samples

⑥ **Interspecies Interactions and Metabolic Profiling**

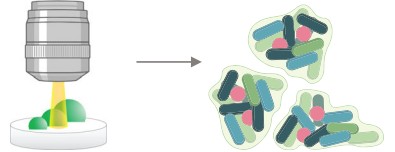

- Acidogenicity/aciduricity/metabolism
- Species prioritization for biofilm studies

⑦ **Biofilm Real-time Imaging and Biogeography**

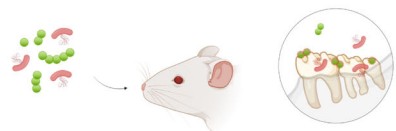

- Spatial structuring
- Spatiotemporal dynamics
- Computational imaging

⑧ **In vivo Virulence Studies**

- Bacterial colonization on teeth
- Disease onset/severity

**Fig. 1 | Overview of methods employed in the study.** Overview of the design of the multi-method, discovery-validation experimental approach employed in the present investigation.

availability of clinical isolates. Based on these criteria, from the list of 16 significant species we prioritized *S. mutans* (the known, well-established pathogen), and *S. sputigena*, *P. salivae*, and *L. wadei* (as new candidates). These four taxa, hereafter referred to as "top species", were carried forward to in vitro virulence assessments and biofilm characterization and served as candidates for further in vivo colonization and pathogenicity studies.

### Interspecies metabolic interactions and acidogenesis

To form the acidic microenvironment that causes demineralization of tooth surfaces, bacterial pathogens within the biofilm microbiota must efficiently metabolize dietary sugars such as sucrose (the primary sugar associated with dental decay) and produce acids (acidogenicity) while surviving in an acidified milieu (aciduricity)[24]. We first sought to determine whether the top species fit the pathogenic profile of being both acidogenic and aciduric.

Glycolytic pH-drop assays for each species individually (Fig. 4A1) and the new candidates combined with the established *S. mutans* (Fig. 4A2) showed that all were active acid producers in the presence of sucrose, lowered pH to highly acidic values, i.e., pH 4.3 to 5.5 which can cause enamel demineralization[25], consistent with up-regulation of glycolysis in the disease-associated plaque biofilm transcriptome. *S. mutans* was the most acidogenic, followed by *L. wadei*, *P. salivae*, and *S. sputigena* (Fig. 4A1). Intriguingly, the combination of *S. mutans* and *S. sputigena* resulted in the highest average acid production rate, significantly higher than that of *S. mutans* alone (Fig. 4A2) and the most rapid pH drop. These findings are suggestive of a possible co-metabolic relationship between *S. mutans* and *S. sputigena*.

All 4 species could grow in an acidic culture medium (i.e., pH pre-adjusted to 5.5), demonstrative of their ability to tolerate acidic conditions required for the development of caries lesions. Acid tolerance challenges of monocultures showed that all species could grow at an

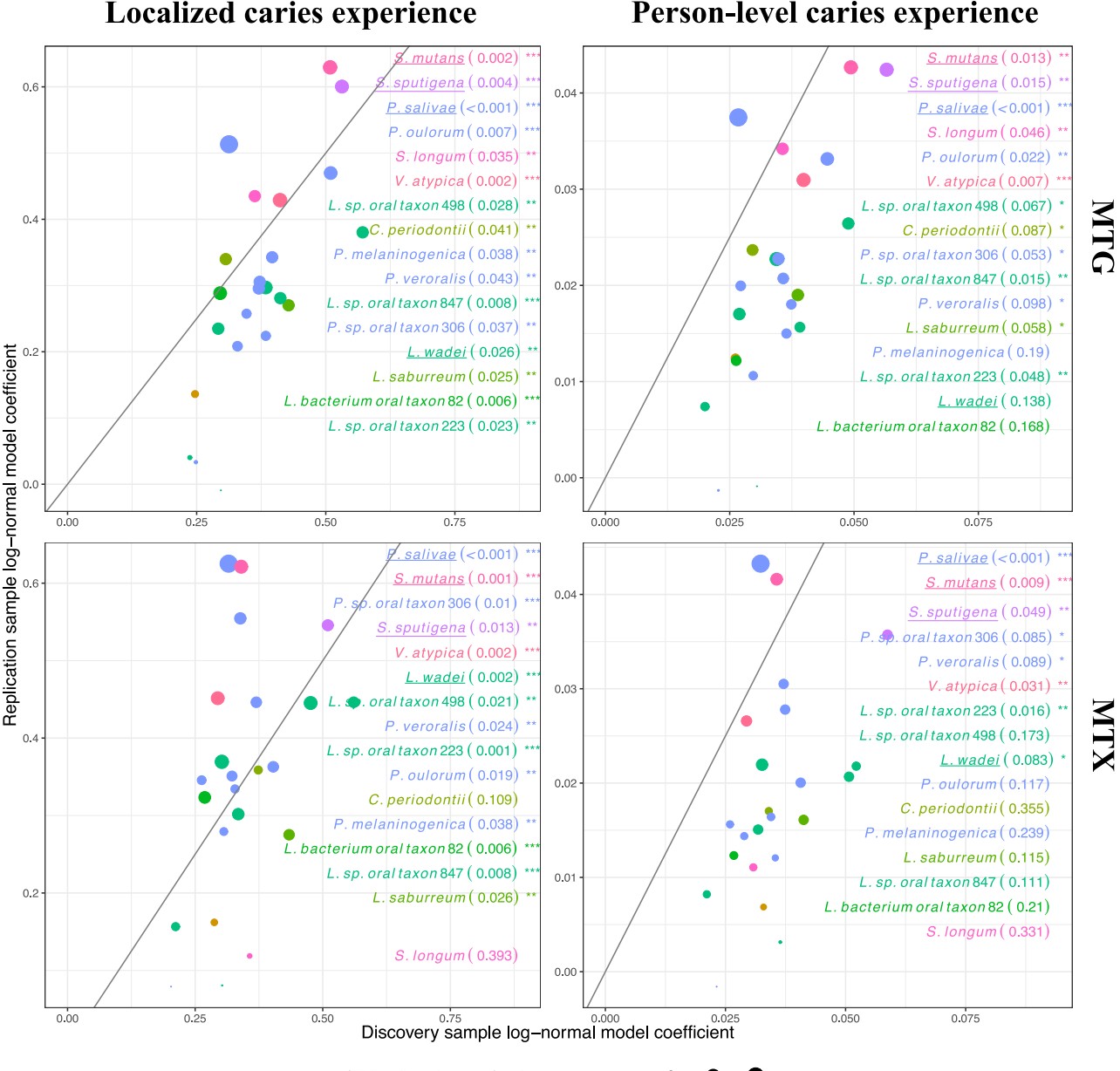

**Fig. 2 | Association of oral microbial species with dental caries.** Estimates of association between dental caries experience and the abundance of 23 species that were significantly associated with caries experience in MTG and MTX analyses in the discovery ($n = 300$ children) sample. The magnitude of association was measured by log-normal model coefficients (the difference of the log-abundance level of each species between the binary caries statuses), and two-sided Wald tests were performed. 16 of those species showed evidence of association in the replication sample ($n = 116$ children) and are annotated by name in the figure. Nominal p-values are presented in parentheses, and *, **, and *** denote nominal p-values of less than 0.1, 0.05, 0.01, respectively in the replication sample. The effect sizes and their 95% confidence intervals are available in the source data file in the Supplementary Information. Diagonal lines represent $y = x$. Similarly colored taxa are members of the same genus. The 4 underlined taxa are the "top species" that were prioritized for in vitro virulence assays, biofilm studies, and were candidates for in vivo experiments.

initial pH of 5.5, and further decreased pH to 4.8–4.6 (Fig. 4B1). Notably, acid tolerance was higher when *S. sputigena*, *L. wadei*, and *P. salivae* were in co-cultures with *S. mutans*, demonstrating ability to grow at a starting pH of 5.2 and further decrease the pH to a range between 3.9 to 4.2 after 48 h (Fig. 4B2). Of note, the mixed culture of *S. sputigena* and *S. mutans* achieved the lowest final pH of 3.9 after 48 h.

#### *S. mutans* dominates metabolic activity in top species' mixed cultures

Next, we used real-time isothermal microcalorimetry to characterize the energy (total heat) released by top species' sucrose metabolic activity, individually and when the three new candidates were co-cultured with *S. mutans*. In monocultures, *S. mutans* produced the highest peak and rate of metabolic activity, the fastest times to activity and peak, but also the shortest decay time, reflecting a rapid drop in its metabolic activity (Fig. 4C1). In contrast, *S. sputigena*'s metabolic activity peaked much later (i.e., at 49 h) and decayed slowly, findings aligned with being a slow growing anaerobic bacterium[26], whereas *L. wadei* presented the second highest peak. Co-cultures of *S. sputigena*, *L. wadei* and *P. salivae* with *S. mutans* had metabolic peaks close to the *S. mutans*' monoculture peak (Fig. 4C2) indicating the metabolic dominance by the pathogen within co-cultures.

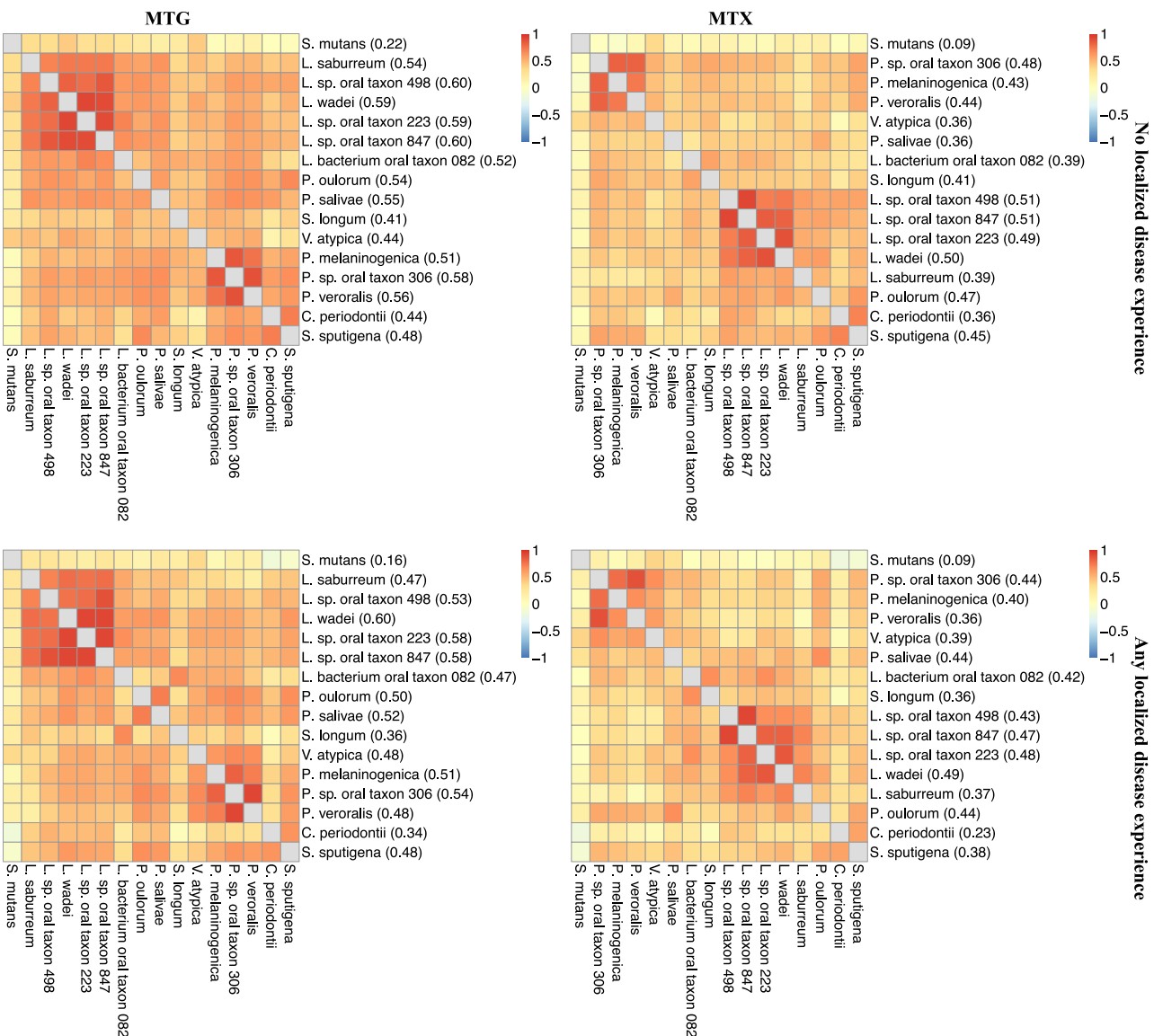

**Fig. 3 | Pairwise correlations between oral microbial species.** Pairwise correlations (i.e., Pearson correlations of residuals generated from a log-normal model for quantitative localized caries experience including terms for batch effects and age) between the 16 significant species' abundance in MTG and MTX, in dental health (i.e., no localized caries experience) and disease (i.e., any localized caries experience). Parenthesized values are individual species' mean correlations with the other 15.

In summary, we found that *S. sputigena*, *L. wadei* and *P. salivae*, alongside the known *S. mutans*, display aciduric and acidogenic properties. All species can metabolize sucrose, the primary disease-associated dietary sugar, produce acids in the glycolytic assay, and survive in the resulting acidic pH. While the metabolic activity is driven by *S. mutans*, we unexpectedly found that the addition of *S. sputigena* results in faster acid production and increased aciduricity, significantly higher than *S. mutans* alone. This suggests a possible cooperative relationship between *S. mutans* and *S. sputigena* that favors the formation of pathogenic biofilms.

### *S. mutans* enhances *S. sputigena* colonization and biofilm formation

Dental caries is a prime example of a biofilm-mediated disease that is initiated by colonization of microbes on tooth surfaces and development of structured biofilm communities[27]. We developed an experimental model using saliva-coated hydroxyapatite discs (sHA, a tooth enamel surrogate) and confocal live-cell imaging to investigate the four top species' biofilm-forming abilities. Because *S. mutans* is a well-characterized pathogen known for its exceptional ability to assemble biofilms on teeth[28], we focused on investigating biofilm forming properties of *L. wadei*, *P. salivae*, and *S. sputigena* alone and in combination with *S. mutans*. We found that the new candidates could co-colonize the sHA surface, with *S. mutans* forming structured but morphologically distinct mixed-species biofilms after 24 h (Fig. 5A upper panel). In contrast, none of the new species alone could efficiently colonize the surface and develop biofilms, with only single cells (*L. wadei* and *P. salivae*) or small bacterial clusters (*S. sputigena*) on the surface (Fig. 5B). Higher abundances of each of the new species were noted when grown as mixed biofilms with *S. mutans* (Fig. 5A lower panel). These findings were further corroborated by quantifying the surface-bound cell volume (Fig. 5A lower panel and Fig. 5B) using a computational image analysis toolbox optimized for biofilms[29]. Indeed, we confirmed higher amounts of *L. wadei*, *P. salivae*, and *S. sputigena* on the hydroxyapatite surface after 24 h when co-cultured with *S. mutans*

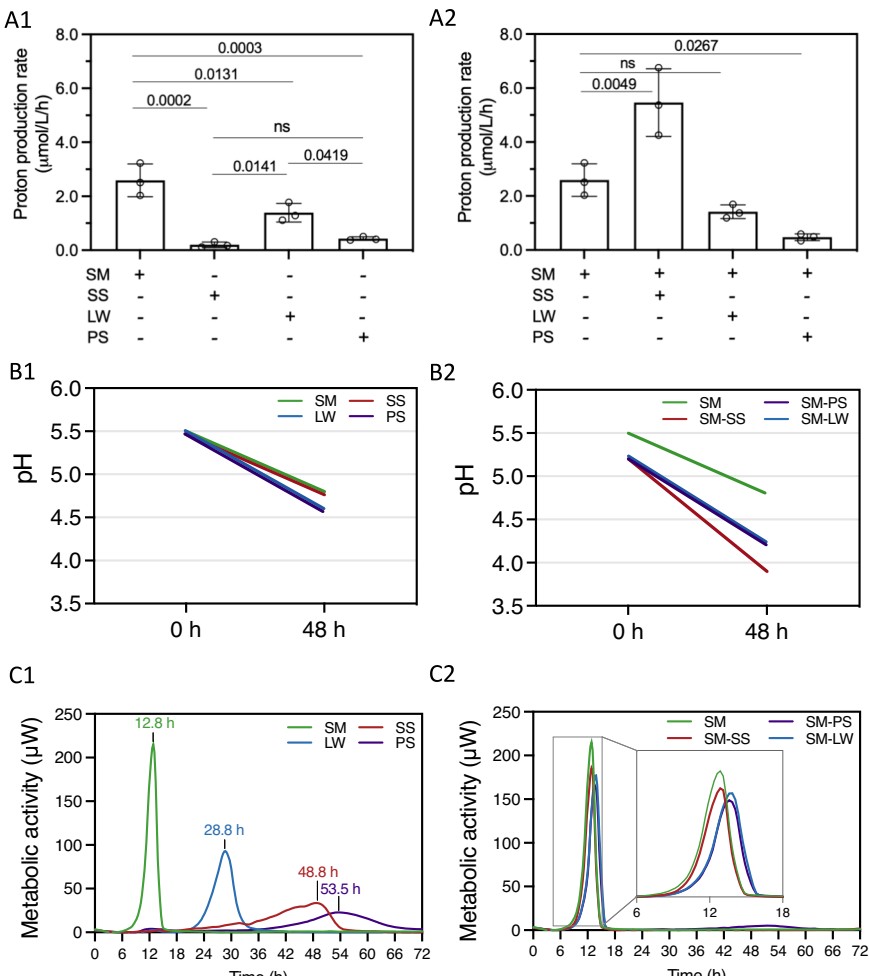

**Fig. 4 | Metabolic and acidogenic profile of the top species.** Top species include *S. mutans*, *S. sputigena*, *L. wadei*, and *P. salivae*. Each species was grown alone or in combination with *S. mutans*. **A1** Average proton production rate of mono-species culture over a 12-h glycolytic assay. **A2** Average proton production rate of mixed-species cultures (*S. mutans* + new species) and *S. mutans* alone over a 12-h glycolytic assay. **B1** Acid tolerance presented as the lowest starting pH that each of the species could survive and the final pH (after 48 h) **B2** lowest starting pH that each of the mixed-species could survive, and the final pH (after 48 h). **C1** Curves of the

metabolic activity (µW) of mono-species cultures. Data above individual peaks indicate the peak metabolic activity time of each species. **C2** Curves of the metabolic activity (µW) of mixed-species cultures (*S. mutans* + new species) and that of *S. mutans* alone. Inset, metabolic curves between 6–18 h. SM, *S. mutans*; SS, *S. sputigena*; LW, *L. wadei*; PS, *P. salivae*. For **A1** and **A2**, data are plotted as mean ± standard deviation from three independent experiments and significant *p*-values are noted above the bars ($p < 0.05$ derived by one-way ANOVA with post hoc Tukey HSD test). ns denotes differences not statistically significant ($p > 0.05$).

---

compared to the same species alone (Fig. 5C), indicating that the presence of *S. mutans* can facilitate their surface colonization. Notably, we found that surface-colonized *S. sputigena* displayed the highest (>10-fold) increase when co-cultured with *S. mutans*, which was higher than that of *L. wadei* or *P. salivae*, suggesting that interspecies interactions may significantly enhance the colonization and biofilm formation by *S. sputigena*.

To further understand the growth dynamics of the single and mixed biofilms, we assessed their attachment at an initial stage (i.e., at 4 h) and at an intermediate stage (i.e., at 10 h). The data indicate that *S. sputigena*, *L. wadei*, and *P. salivae* all bind similarly to the surface either alone or co-cultured with *S. mutans* at 4 h (Supplementary Fig. 3, upper). Interestingly, when co-cultured, *S. sputigena* appears to form clusters near *S. mutans* aggregates in the mixed biofilm at 10 h (Supplementary Fig. 3, middle, white arrowheads), suggesting early physical interactions between the two species during biofilm initiation. These observations suggest that the post-binding events, i.e., close interactions and co-metabolism with *S. mutans*, may mediate the biofilm spatial structure and community development at later stages, i.e., at 24 h (Supplementary Fig. 3, lower).

## Interspecies spatial structuring and co-localization within biofilms

Given that *S. mutans* produces extracellular polysaccharides (EPS) that enhance bacterial cell co-adhesion and biofilm accumulation[30], we investigated the spatial localization of the three new candidate species, *S. mutans*, and EPS. High-resolution images of magnified areas showed that most of *L. wadei*, *P. salivae*, and *S. sputigena* cells were spatially located in between streptococcal clusters (known as microcolonies), rather than intermixing with the *S. mutans* cells within (Fig. 5D, upper panel). Interestingly, we found a densely packed accumulation of *S. sputigena*, whereas *L. wadei* or *P. salivae* sparsely colonized these areas. Using a fluorescent marker that specifically labels *S. mutans*-derived EPS α-glucans[28], we found that most of *L. wadei*, *P. salivae*, or *S. sputigena* cells were co-localized with the extracellular polymeric matrix (Fig. 5D, lower panel).

We performed three-dimensional analyses to characterize the co-localization of two fluorescent signals in relation to each other using Mander's overlap coefficient, a quantitative measure of spatial proximity[31]. For each mixed-species biofilm, we calculated two Manders' coefficients: 1) between each new species and *S. mutans* and 2)

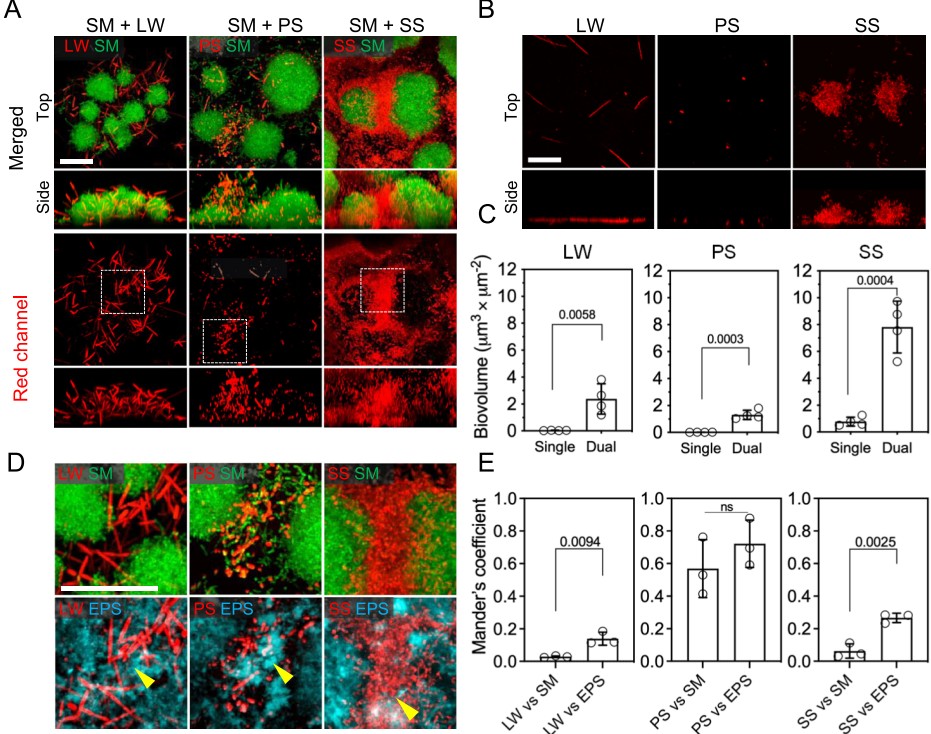

**Fig. 5 | Biofilm formation and spatial structuring on tooth-mimetic surface.**
**A** Confocal images (top and side views) of 24-h mixed-species biofilms on saliva-coated hydroxyapatite surfaces formed by each of the new species together with *S. mutans*. The upper panel is a merged image demonstrating the spatial structuring of *S. mutans* (in green) and the new species (in red) within mixed biofilms. Lower panel, red channel only. Dotted box, areas that are shown magnified in panel **D**.
**B** 24-h mono-species biofilms formed by the top species (top and side views)
**C** Biovolume of the new species within single and mixed biofilms, based on

computational image analysis of panels **A** and **B**. **D** Top, magnified confocal images of mixed species biofilms; bottom, the new species physically interacting with extracellular α-glucan matrix (EPS, in cyan). **E** Computational colocalization analysis of the new species versus *S. mutans* cells or EPS. SM, *S. mutans*; SS, *S. sputigena*; LW, *L. wadei*; PS, *P. salivae*. Scale bars, 20 μm. For **C** and **E**, data are plotted as mean ± standard deviation from three independent experiments, and significant *p*-values are noted above the bars (*p* < 0.05 derived from two-sided Student's *t*-test). ns denotes differences not statistically significant (*p* > 0.05).

between each new species and EPS. We found that *L. wadei* and *S. sputigena* were more proximal to EPS than *S. mutans* (Fig. 5E, left and right panels), indicating that these species may be primarily associated with EPS than to *S. mutans* cells. In contrast, *P. salivae* showed similar proximity to *S. mutans* cells and EPS (Fig. 5E, middle panel). In summary, our findings show that the new candidates form mixed biofilms with *S. mutans*, and this coexistence enhances their surface colonization and co-development of structured biofilms. *S. sputigena* forms a densely populated community interspersed between *S. mutans* clusters, leading to highly cohesive, co-assembled biofilms.

## The motile bacterium *S. sputigena* becomes trapped in streptococcal exoglucans

Motivated by the enhanced surface colonization of *S. sputigena* in the presence of *S. mutans* and the enhanced acidogenicity of this species combination, we sought to characterize the spatial structuring of *S. sputigena* within the mixed-species biofilm. As a backdrop, *S. sputigena* was originally found in the aerodigestive system (including saliva), the subgingival microbiota of patients affected with periodontitis[32–34], and endodontic infections[35]. *Selenomonas* multispecies communities were common (i.e., among the top 10 abundant taxa) in the biofilms obtained in our recent in vivo experimental study of caries progression and arrest[36]. Its role in supragingival biofilms and how it interacts with other microbes remain unknown.

Unlike most oral microbes, *S. sputigena* has flagella, which are surface-attached appendages that allow motility in liquid environments. Unexpectedly, we found that *S. sputigena* cells were motile even after their colonization on the sHA surface, displaying a tumbling, multi-directional motion (Supplementary Movie 1) while remaining

attached to the hydroxyapatite surface. We used real-time live imaging and computational motion tracking to study the surface motility of *S. sputigena* cells when co-cultured with *S. mutans* (Fig. 6A, B and Supplementary Movie 2). The spatial coordinates of individual *S. sputigena* cells were followed at each time frame and a time-resolved trajectory was generated (representative frames shown in Fig. 6B). Our results revealed distinctive motility behaviors of *S. sputigena* cells depending on their spatial location relative to that of *S. mutans*. The upper panels of Fig. 6C (same areas as "Box a", in Fig. 6A) show that *S. sputigena* cells near *S. mutans* cells display no motility (Fig. 6C top). In contrast, *S. sputigena* cells located far away from *S. mutans* remained motile (lower panels of Fig. 6C; same areas as "Box b" in Fig. 6A). We also calculated accumulated displacements (total path length) of individual *S. sputigena* cells relative to their original position. The displacement curves revealed that most *S. sputigena* cells adjacent to *S. mutans* did not move or only moved for short distances (Fig. 6D, left panel), whereas those with no *S. mutans* nearby moved actively for longer distances (Fig. 6D, right panel).

Given the contrasting motility between *S. sputigena* cells at different spatial locations, we hypothesized that *S. sputigena* cells that lost their surface motility might be physically trapped in *S. mutans*-derived EPS. We first tracked *S. sputigena* cells that were co-localized with EPS α-glucans and found that the cells were devoid of mobility (Fig. 6E, upper panels). Then, we performed EPS degradation using glucanohydrolases (mutanase and dextranase) that specifically hydrolyze α-glucans without antibacterial activity[37] and examined the motility behavior of *S. sputigena* in real-time. We found that the surface motility of *S. sputigena* was recovered following α-glucans degradation (trajectories shown in Fig. 6E, lower panels). These observations were confirmed by

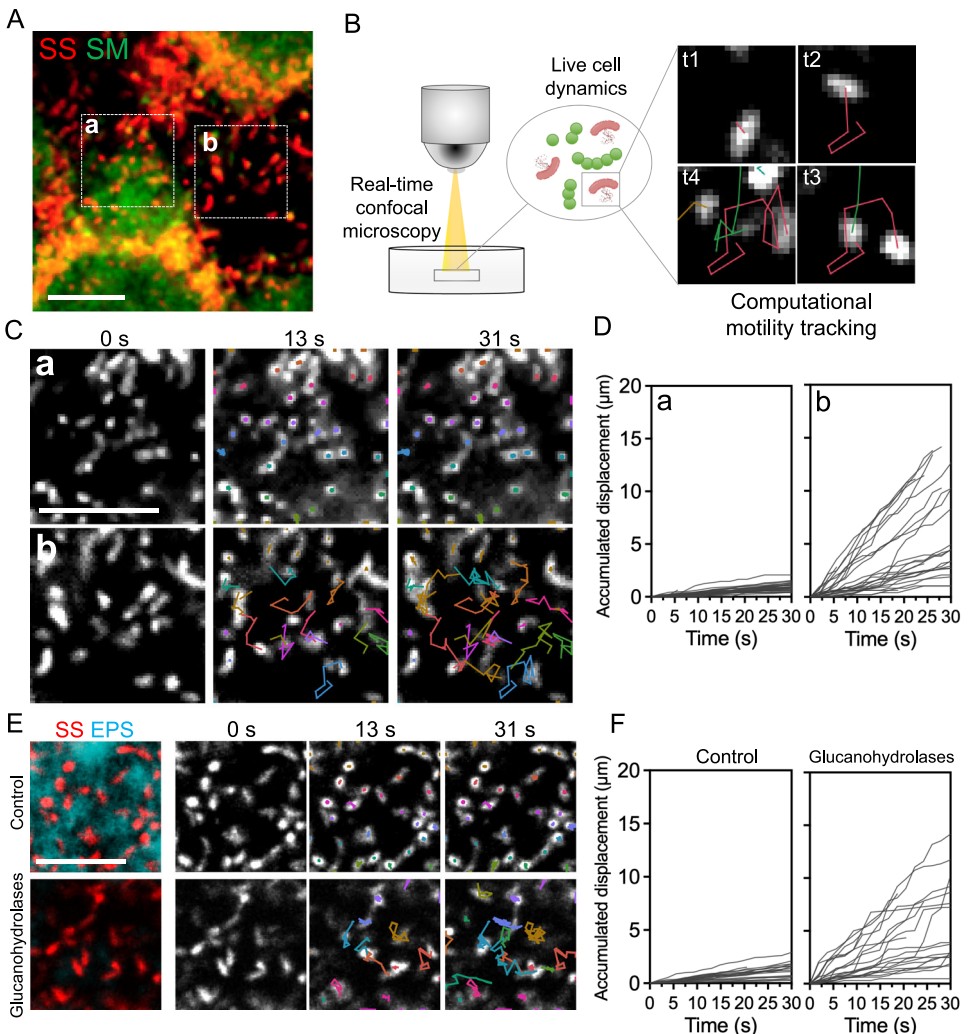

**Fig. 6 | _S. sputigena_, a motile bacterium, becomes trapped in extracellular α-glucans. A** Confocal image of _S. sputigena_-_S. mutans_ mixed biofilm formed on the surface. Red, _S. sputigena_; green, _S. mutans_. Dotted boxes, biofilm subareas with _S. sputigena_ physically associated with _S. mutans_ clusters (**a**) or surface-attached _S. sputigena_ alone (**b**). **B** Surface-attached microbes were visualized by fluorescence staining and real-time confocal microscopy. Individual _S. sputigena_ cells (red channel in panel **A**) were tracked computationally over time to generate spatio-temporal trajectories. **C** Trajectories of surface-attached _S. sputigena_ cells in Areas (**a**) and (**b**), as shown in panel **A**. Colors indicate trajectories that originated from individual cells. **D** Accumulated _S. sputigena_ cell displacement (total path length)

relative to the initial position. Left, accumulated displacement of _S. sputigena_ cells in Area (**a**); right, in Area (**b**). **E** Immobilized _S. sputigena_ cells trapped by _S. mutans_-derived α-glucan matrix. Red, _S. sputigena_; cyan, α-glucan matrix. Top panel, _S. sputigena_ cells trapped by α-glucans showed no mobility; bottom panel, upon EPS degradation using glucanohydrolases (dextranase and mutanase), surface mobility of _S. sputigena_ cells is recovered. **F** Accumulated _S. sputigena_ cell displacement (total path length) relative to the initial position. Left, accumulated displacement of _S. sputigena_ cells in the control group; right, after EPS enzymatic degradation. SM, _S. mutans_; SS, _S. sputigena_. Representative images from three independent experiments are shown. Scale bars, 10 μm.

computational motility tracking before and after glucanohydrolase treatment (Fig. 6F), indicating that EPS degradation released trapped _S. sputigena_ cells restoring their surface motility. Additionally, _S. sputigena_ single-species biofilms were formed in the presence of exogenously supplemented purified glucosyltransferase B (GtfB), an _S. mutans_-derived exoenzyme that produces α-glucans[30]. _S. sputigena_ cells grown with GtfB (without _S. mutans_) were trapped by α-glucans produced by the enzyme, showing no surface motility whereas most _S. sputigena_ cells growing without Gtf were motile (trajectories in Supplementary Fig. 4A; displacement curves in Supplementary Fig. 4B), further supporting EPS-mediated entrapment and loss of motility.

By co-colonizing the surface with _S. mutans_, _S. sputigena_ cells appear to lose motility and become immobilized through interspecies cell-glucan matrix interactions yet remain viable and active, which may enhance surface binding and accumulation of these motile bacteria that could influence the scaffolding of the mixed biofilm community. Our biofilm MTX data corroborate these observations. First, we found that _S._

_mutans'_ _gtfC_ gene (encoding glucosyltransferase GtfC, involved in glucan synthesis) and the glucose-1-phosphate adenylyltransferase gene were significantly upregulated in disease (Supplementary Table 3). _gtfC_ encodes an enzyme that synthesizes both soluble and insoluble α-glucans[30] that are highly susceptible to degradation by dextranase (breaks down soluble glucans) and mutanase (digests insoluble glucans) which were used in this study's motility experiments. Second, we found several negative _S. mutans_ and _S. sputigena_ gene-gene expression interactions. Intriguingly, 11 of 13 gene-gene interactions were negative, and _S. mutans'_ glucose-1-phosphate adenylyltransferase gene demonstrated a strong negative interaction ($p = 3.7 \times 10^{-6}$) with _S. sputigena's_ motility-related flagellin gene (Supplementary Table 4).

**Spatial arrangement of _S. sputigena_ and _S. mutans_ within intact biofilm structure**

Considering that _S. mutans_ can significantly enhance _S. sputigena_ colonization to the tooth-mimetic surface and modulate its surface

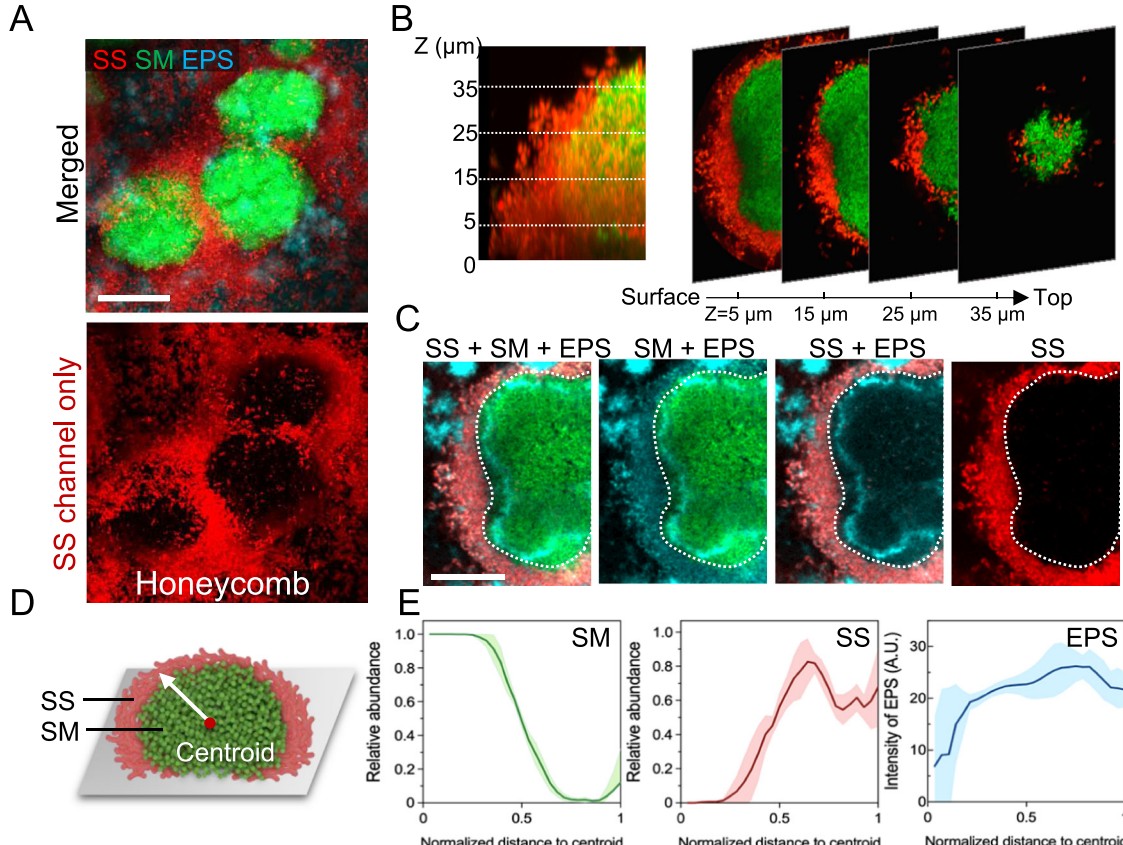

**Fig. 7 | Biogeography of *S. sputigena*-*S. mutans* biofilm. A** The mixed biofilm featured by a honeycomb-like structure formed by *S. sputigena* (in red) encapsulating cell clusters of *S. mutans*. Both species are embedded by EPS α-glucan matrix (in cyan) **B** Optical sectioning of the mixed-species community reveals physical segregation between *S. mutans* (green) and *S. sputigena* (red) within the biofilm. **C** Cross-sectional confocal image of the spatial arrangement of *S. sputigena*, *S. mutans* and EPS. *S. mutans*-derived EPS (cyan) is present both inside and outside of the *S. mutans* inner core cluster (green); *S. sputigena* cells (red) are predominantly outside, embedded in EPS while encapsulating *S. mutans* cluster **D** A diagram of the biogeography and the centroid of mass used as reference for spatial analysis. **E** Spatial distribution of *S. sputigena*, *S. mutans* and EPS relative to the distance to the centroid. Lines correspond to means and shaded regions to standard deviations from three independent experiments. Representative images from three independent experiments are shown. SM, *S. mutans*; SS, *S. sputigena*. Scale bars, 20 µm.

motility, we investigated how this co-colonizing community mediates biofilm spatial structure (biogeography) at multi-length scales. To this end, we used super-resolution confocal microscopy coupled with computational analyses for mixed biofilm communities[38]. We found a unique 3D multicellular organization composed of densely packed *S. sputigena* cells encasing *S. mutans* microcolonies (Fig. 7A). Using layer-by-layer imaging sectioning, we found an orderly arrangement of *S. sputigena* cells across the entire height of the biostructure (Fig. 7B). A representative sectional rendering (z = 15 µm from the surface) of the 3D biostructure revealed a spatially segregated inner core formed primarily by *S. mutans*, and a dense ring-like outer layer of *S. sputigena*, both associated with EPS α-glucans (Fig. 7C). We investigated the composition and the spatial structuring of the microbial and EPS components in relation to the center of mass (referred to as centroid) of the biostructure (see diagram in Fig. 7D). We found that the core near the centroid consisted predominantly of *S. mutans*, whereas the periphery harbored predominantly *S. sputigena* (Fig. 7E, left and middle panels). Notably, *S. mutans*-derived EPS α-glucans were detected throughout the entire biostructure, including the peripheral areas where the abundance of *S. mutans* cells was low (Fig. 7E, right panel), suggesting that the secreted EPS could mediate co-adhesion without direct cellular contact providing scaffolding for the interspecies assembly.

To further assess the structural organization of the biofilm, we performed 3D reconstruction of the confocal images and quantitative analysis of the spatial distribution of the bacterial cells. This analysis

revealed a unique honeycomb-like architecture consisting of *S. sputigena* cells forming a multicellular superstructure (Fig. 8A and Supplementary Movie 3). To determine the spatial distribution of bacterial cells, we computationally dissected the large mixed-species biostructure into small cubic volumes in micron scale, and calculated the relative abundance of (i.e., the proportion of the total cellular volume occupied by) *S. sputigena* cells within each cube in correspondence to their 3D spatial coordinates[29]. Using this method which provides local distribution with 3D spatial resolution, we found high *S. sputigena* abundance near the entire outer surface of the biostructure (selected horizontal or vertical planes shown in Fig. 8B). These results indicate a non-random pattern with densely packed yet spatially segregated biogeography between *S. sputigena* and *S. mutans* cells. Taken together, we find evidence for a motile bacterium, *S. sputigena*, becoming immobilized by a foreign EPS produced by a co-colonizing species (*S. mutans*), promoting localized growth and accumulation, and forming a distinctive multicellular honeycomb superstructure guided by the matrix scaffolding.

## *S. sputigena*-mediated interspecies spatial structuring is required and specific for enhanced biofilm virulence

To investigate whether spatial structuring is required for the biofilm to create acidic microenvironments that are critical for dental caries development, we measured the biofilm-apatite interface pH in situ. We used a pH-responsive fluorescent probe[39] and high-resolution confocal live imaging to measure pH changes in real-time. We found rapid

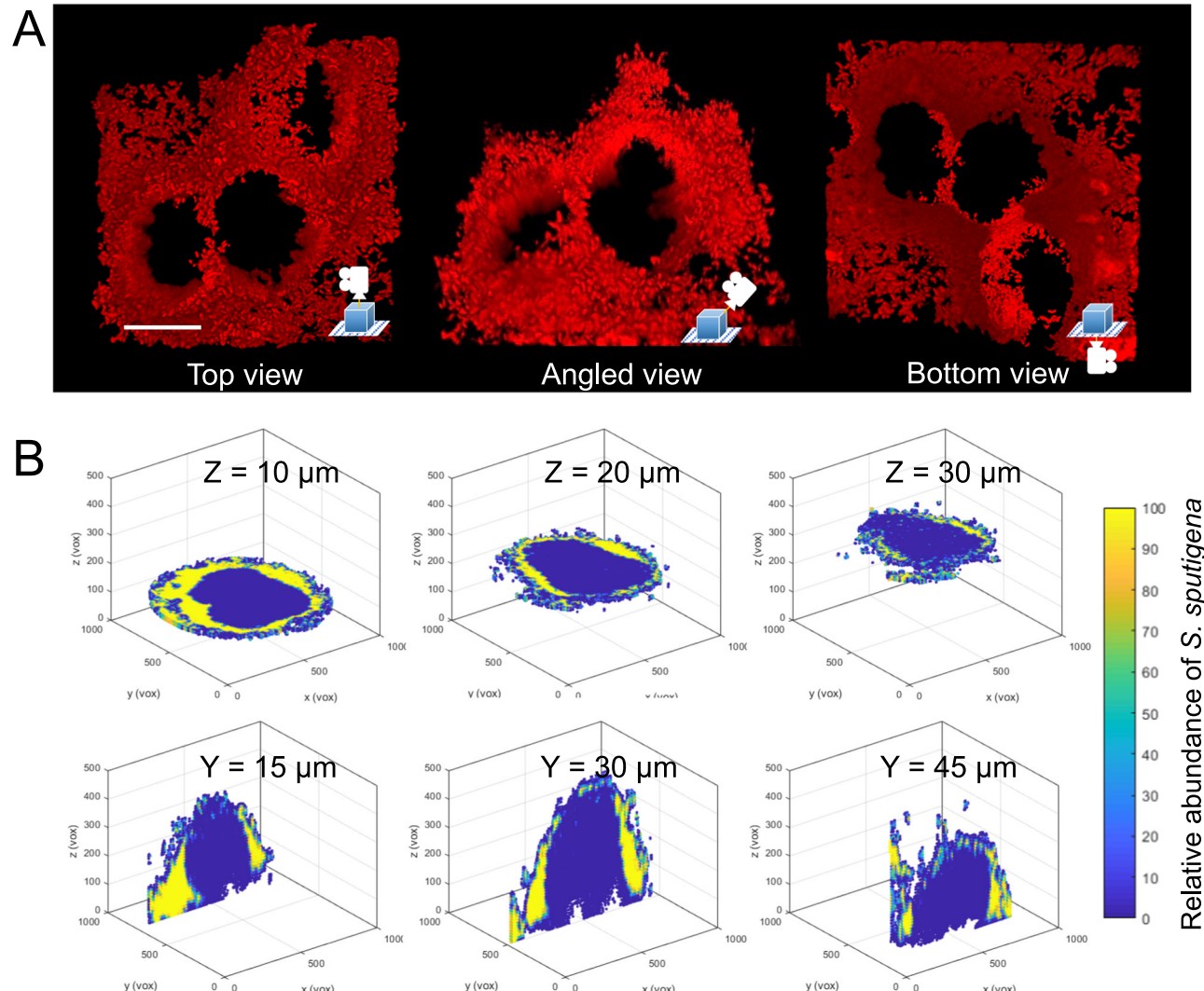

**Fig. 8 | Spatial distribution of *S. sputigena* within the biofilm superstructure.** **A** Three-dimensional rendering of the honeycomb structure formed by *S. sputigena* in the *S. sputigena-S. mutans* mixed biofilm. **B** Spatial distribution of *S. sputigena* (relative abundance, defined as the proportion of the total cellular volume occupied by *S. sputigena*) within the community structure at different horizontal (top) and vertical planes (bottom). Scale bar, 20 μm.

acidification at the biofilm-apatite interface of the superstructure formed by *S. sputigena* and *S. mutans* after glucose exposure, which resulted in highly acidic (pH < 5.5) regions (Supplementary Fig. 5A, upper). Given that *S. mutans*-derived EPS α-glucans were detected throughout the entire biostructure mediating co-adhesion for the interspecies assembly (Fig. 7C, E), we used a *S. mutans* double-knockout of *gtfB* and *gtfC* (i.e., defective in α-glucans production[30]) to disrupt the biofilm spatial structuring with *S. sputigena*. We found that *S. sputigena* and *S. mutans* Δ*gtfBC* mutants were unable to form structured biofilms and did not result in acidic pH regions (Supplementary Fig. 5A, middle). This finding was further corroborated by co-culturing *S. sputigena* and wild type *S. mutans* in the presence of glucanohydrolases (dextranase and mutanase) that specifically break down α-glucans and dismantle the biofilm structure without affecting bacterial viability, impairing the ability to create localized acidic-pH regions (Supplementary Fig. 5A, lower). The data suggest that the interspecies spatial structuring between *S. sputigena* and *S. mutans* is required to create a virulent acidic microenvironment.

We further investigated the specificity of *S. sputigena-S. mutans* interactions to promote biofilm virulence using an ex vivo human tooth-enamel biofilm model. This method allows direct assessment of the acid-induced damage (i.e., demineralization) on the enamel surface inflicted by the biofilm[40]. Given that *P. salivae* does not increase the proton production rate when co-cultured with *S. mutans* (Fig. 4, A2), we used a *P. salivae-S. mutans* co-culture as control and compared to the acid-induced enamel demineralization caused by *S. sputigena-S. mutans* biofilms, as well as by *S. mutans* alone. Macroscopically, we found large areas of enamel demineralization associated with the *S. mutans-S. sputigena* biofilm, which was characterized by chalky-like opaque surfaces under the stereoscope (Supplementary Fig. 6A, lower left), similar to caries lesions found clinically. In contrast, only small areas of opaque demineralized areas were found on the enamel surface from *P. salivae-S. mutans* (Supplementary Fig. 6A, middle left) and *S. mutans* alone (Supplementary Fig. 6A; upper left) biofilms. These differences were confirmed using confocal topography imaging and transverse microradiography analysis.

The enamel surfaces under *S. mutans-S. sputigena* biofilms showed higher roughness compared to those from *S. mutans* biofilms and *P. salivae-S. mutans* biofilms (Supplementary Fig. 6C, "Roughness of enamel surface"; *p* < 0.05), with widespread regions of surface damage (Supplementary Fig. 6A, lower right). In contrast, the surface roughness from *P. salivae-S. mutans* biofilms was similar to that from *S. mutans* alone (Supplementary Fig. 6C; *p* > 0.05), both showing milder enamel surface demineralization (Supplementary Fig. 6A right).

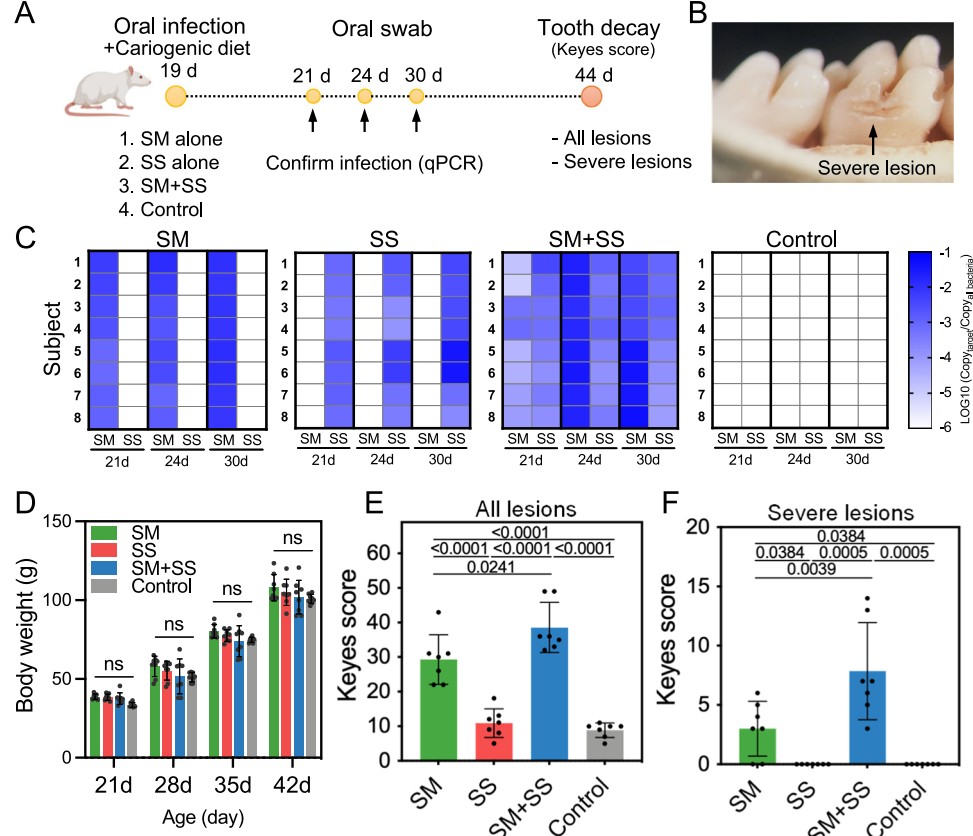

**Fig. 9 | In vivo study of the role of *S. sputigena* and *S. mutans* in dental caries.** **A** A diagram of the experimental design. **B** Cavitated (severe) carious lesions developed on the animal's teeth, similar to those found clinically in severe childhood tooth decay. **C** Confirmation of infection by qPCR. Oral swabs were taken on Day 21, Day 24, and Day 30 and were subject to qPCR analysis using species-specific probes. **D** Body weight of the animals was measured weekly to monitor the systematic impact of the bacterial infection on animal health during the experiment period. No significant differences were observed between groups. **E** Keyes scoring of total caries (tooth decay) developed on the smooth surfaces. **F** Keyes scoring of cavitated (severe) lesions developed on smooth surfaces. Caries scores were recorded as stages and extent of carious lesion severity according to Larson's modification of Keyes' scoring system. Data are presented as mean ± standard deviation. ($n = 8$ animals) and significant $p$-values are provided above the bars ($p < 0.05$, derived from one-way ANOVA with post hoc Tukey HSD test). ns denotes differences not statistically significant ($p > 0.05$).

Transverse microradiography analysis validated the extent of demineralized enamel lesion, showing significantly higher mineral loss and deeper acid-damage caused by *S. mutans-S. sputigena* biofilms ($p < 0.05$) versus *P. salivae-S. mutans* or *S. mutans* biofilms (Supplementary Fig. 6B, C). Taken together, the data suggest specificity of *S. sputigena-S. mutans* interactions and spatial structuring to promote biofilm virulence.

### *S. sputigena* can colonize tooth surfaces and increases caries experience in vivo

The in vitro and ex vivo experimental data presented thus far demonstrate that *S. sputigena* can bind and form highly structured, densely packed biofilms with *S. mutans* on apatitic surface—their combination produces more acid and more enamel demineralization than either species alone. It is thus logical to expect that they can co-colonize tooth surfaces and exhibit enhanced biofilm virulence in vivo. To experimentally test this hypothesis, we sought to determine whether *S. sputigena* alone or when co-infected with *S. mutans* (Fig. 9A) can colonize tooth surfaces and cause tooth decay based on an established rodent model that mimics the characteristics of early childhood caries, including exposure to a sugar-rich diet and rampant dental cavitation[41] (Fig. 9B). We found that the animals were infected by *S. mutans*, *S. sputigena*, or both and remained persistently infected (Fig. 9C) as determined by real-time PCR using species-specific probes. The uninfected (control) animals remained free of infection by *S. mutans* or *S.*

*sputigena* (Fig. 9C). All animals maintained good health showing a steady body weight gain with no significant difference between groups (Fig. 9D). These data confirm the ability of *S. sputigena* to co-colonize tooth enamel surfaces with *S. mutans* in vivo.

The impact on caries development and severity was then assessed for each experimental condition. In this model, tooth enamel progressively develops caries lesions (analogous to those observed in humans), proceeding from initial areas of demineralization to moderate lesions and on to extensive (i.e., severe) lesions characterized by enamel structure damage and cavitation. Consistent with previous studies[42,43], *S. mutans* infection led to the onset of dental caries whereas control uninfected animals harboring their natural oral microbiota developed minor tooth demineralization with no severe lesions (Fig. 9E). Infection with *S. sputigena* did not induce development of caries lesions on the tooth surface compared to the uninfected control (Fig. 9E) despite confirmed bacterial colonization, suggesting that, alone, *S. sputigena* has limited cariogenic potential. However, *S. sputigena* co-infected with *S. mutans* resulted in a significant increase of both total caries experience and the severity of caries lesions compared to those infected with either species alone (Fig. 9E), indicating enhanced virulence within resident microbiota. Notably, we found that co-infection produced more severe lesions (Fig. 9F), characterized by enamel destruction and frank cavities (Fig. 9B). In sum, we show that *S. sputigena* interacts with *S. mutans* for enhanced co-colonization and the presence of *S. sputigena* leads to the assembly of

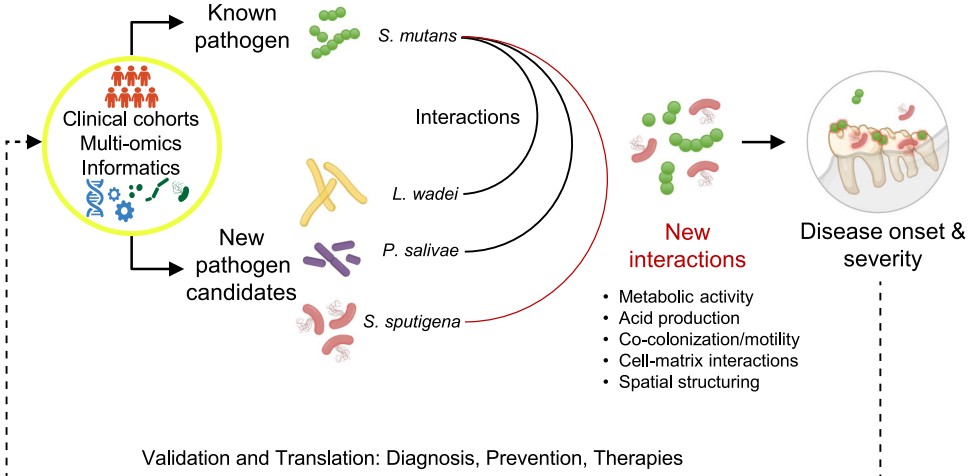

**Fig. 10 | Application of the multi-method, discovery-validation pipeline in the present study.** Outline of the study's experimental approach that can serve as a guide for future oral microbiome discovery-validation investigations. We began by analyzing and integrating multi-omics data generated from human plaque samples obtained in two community-based clinical samples of preschool-age children. Then, we carried forward initial discovery results into a validation pipeline combining informatics, laboratory, and in vivo experimental models seeking to identify pathogen candidates and obtain insights into how these previously uncharacterized species interact and form virulent oral biofilms.

uniquely structured biofilms with enhanced virulence that cause the onset of severe dental disease in vivo, suggesting a new pathobiont and interspecies biogeography to promote disease-causing conditions.

## Discussion

In this study, we integrated multi-omics of human plaque samples from two community-based clinical cohorts in a discovery-validation pipeline combining informatics, laboratory, and in vivo experimental models to identify pathogen candidates and obtain insights into how these previously uncharacterized species interact and form virulent oral biofilms (Fig. 10). Comprehensive taxonomic association analyses and multiple testing-controlled, across-clinical trait criteria identified 16 species significantly associated with early childhood caries, including several new candidates. Among them, *S. mutans* (an established pathogen), *S. sputigena*, *P. salivae*, and *L. wadei* (as new candidates) were strongly associated with disease and were carried forward into comprehensive biofilm studies. We found that the flagellated and motile *S. sputigena* develops a cooperative interaction with *S. mutans* to exacerbate biofilm virulence. Specifically, *S. sputigena* and *S. mutans* can co-colonize apatitic surfaces, metabolize sugars, and produce and tolerate acids. Together, the two species contribute to significantly enhanced acidogenesis and aciduricity. This interaction is underlaid by *S. sputigena*'s entrapment in streptococcal exoglucans, wherein it loses motility, and builds a densely packed honeycomb-like multicellular structure. Negative interactions between *S. mutans*' glucose-1-phosphate adenylyltransferase gene and several *S. sputigena*'s genes including the flagellin gene[44] in MTX, corroborate the inter-species interactions manifested in the experimental studies. Crucially, while incapable of causing disease on its own, *S. sputigena* significantly increases the biofilm virulence in vivo when co-infected with *S. mutans* and augments the severity of dental caries. *S. sputigena* has so far been implicated in periodontal disease[23,45], endodontic infections[35], and is a member of aerodigestive tract microbiome with emerging prognostic significance in cancer[46,47]. Our findings demonstrate that *S. sputigena* acts as a pathobiont outside its typical habitat to modulate biogeography (i.e., spatial structure), metabolic activity, and pathogenicity of supragingival biofilms in the context of a prevalent childhood disease.

We leveraged robust clinical and multi-omics data generated among sizeable community-based samples of young children and used conservative selection criteria to identify bacterial species associated with disease experience and guide the prioritization of candidates for characterization and experimental validation. The large sample size for a multi-omics study of this kind and the application of strict discovery and validation criteria led to a set of high-confidence species associated with caries experience, including known and novel taxa. While we were able to comprehensively test four of these 16 species, we anticipate that this set of high-confidence candidates will be further studied in the near future by us and others. Biofilm transcriptomics analyses affirmed significantly upregulated metabolic pathways (e.g., glycolysis, pyruvate fermentation, and UDP-N-acetylglucosamine) and bacterial genes (e.g., glucose-1-phosphate adenylyltransferase) in disease. These findings are aligned with metabolic interactions between microbiota and dietary sugars expected to be found in dental caries[24]. Sugars fuel the emergence of dysbiosis by favoring proliferation of species that can adapt to ecological changes and promote acidification of the biofilm microenvironment that underlies destruction of mineralized dental tissues. Enhanced acidogenesis, up-regulation of UDP-N-acetyl-D-glucosamine biosynthesis I, and gene-gene interactions in MTX between the two species strengthen the case for *S. sputigena* being an important species in the caries-associated dental plaque biofilm[36].

A key finding of the present report is that the cooperative interaction between *S. sputigena* and *S. mutans* hinges upon an unexpected phenomenon wherein a motile bacterium that is typically reported as resident of the subgingival (i.e., below gum line) niche is entrapped in an exopolymeric matrix produced by another species residing in the supragingival (i.e., above gum line) area and forms a unique spatial biofilm structure that increases virulence. We show that the dense growth of immobilized *S. sputigena* cells colonizing the apatite surface and surrounding *S. mutans* cell clusters create a cohesively packed community superstructure that cannot be achieved by either species alone. Importantly, we demonstrate that the immobilization of *S. sputigena* and the spatial structuring of the mixed community are associated with the creation of localized regions of highly acidic pH, a hallmark for biofilm virulence in dental caries. The exact mechanisms by which *S. sputigena* loses motility and forms the densely packed honeycomb structures remain unknown although *S. mutans*-derived EPS α-glucans appear to play a key role. *S. sputigena* expresses a heavily glycosylated flagellin, building blocks of flagella[44]. It is possible that the glycans on the surface of *S. sputigena* display glucan-binding properties, interacting with streptococcal exoglucans, and thereby

immobilize the cell. Additionally, fucose is a major monosaccharide constituent of the O-linked glycans in *S. sputigena* flagellin[44], and our recent metabolomics work among the same discovery sample has demonstrated elevated fucose abundance in early childhood caries[48]. Glucosyltransferases (Gtfs) secreted by *S. mutans* can bind to sugar moieties of different polysaccharides on the surface of other microbes and produce EPS α-glucans in situ[49]. It is possible that Gtf can bind to fucose moiety allowing glucan synthesis in situ that could contribute to immobilization. Future studies need to investigate the interactions of Gtf with *S. sputigena*, glucan-fucose interactions, and determine whether these interactions are strain-dependent.

*S. sputigena* has been recalcitrant to genetic manipulation which hinders in-depth molecular studies to further understand its pathogenic roles, such as the functions of flagellin on inter-species biofilm interactions. However, new genetic engineering methodologies involving restriction modification-silent tools[50] may circumvent this limitation. Aside from *S. sputigena*, several new species emerged as prime targets for additional studies. For example, *L. wadei* and *P. salivae*, both strongly associated with caries experience in our discovery and replication cohorts, may play pathogenic roles in the context of polymicrobial biofilms that are not necessarily associated with *S. mutans* and could be investigated in the future. Another candidate is *Lachnoanaerobaculum saburreum*, which is capable of producing acid from glucose, lactose, and sucrose among other sugars[51] and is involved in pyruvate fermentation to acetate and lactate II—one of the four pathways that we found as significantly differentially expressed in disease. Examination of multi-species combinations of these candidates may provide additional insights about the biofilm virulence mechanisms.

A limitation of our current multi-omics pipeline is that it is not optimized for studying inter-kingdom interactions (i.e., viruses and fungi, besides bacteria). For example, the nucleic extraction protocols were not optimal for detecting and analyzing fungi, which are important in the context of childhood caries[52]. This can be overcome with the incorporation of nucleic acid extraction protocols for downstream fungal analyses. At the same time, the detection of phages in MTG and MTX data is a rapidly growing area that is likely to provide high yields in the near future[53]. Another limitation is the cross-sectional nature of our human observational data, which limits the samples' inferential potential regarding temporality and causality—it is likely that some taxonomic and functional changes found in the initial discovery of candidate species in human plaque samples are downstream (i.e., consequences) of established disease. This is where the multi-modality aspect of our study is of the essence—the experimental studies of the pipeline are well-positioned to help clarify the actual pathogenic potential of candidate species. Nevertheless, community-based longitudinal human studies are warranted to enhance both discovery and validation of new species, not only in the presence of dental caries but also in its onset and progression.

In summary, we employed a multimodal pipeline to gain new knowledge about taxonomic and functional features of the childhood oral microbiome in health and disease from two sizeable discovery and replication community-based samples of human plaque biofilms. We discover a novel inter-species interaction and unique biogeography at microscale wherein a motile flagellated species becomes immobilized in the EPS matrix produced by a disease-causing species and proliferates to build a 3D multicellular superstructure with enhanced acidogenesis. We show that the interaction between *S. sputigena* and *S. mutans* augments the severity of dental caries in the presence of resident microbiota in vivo, suggesting a new pathobiont exacerbating biofilm virulence for a common yet unresolved disease. Further understanding of the spatial structuring function and pathobiont-mediated virulence may reveal new mechanisms of biofilm assembly and therapeutic targets, which may be relevant to other polymicrobial

infections where other species are interacting with known pathogens in complex biofilms.

## Methods

### Human microbiome studies: context and sampling

Human microbiome studies were done in the context of ZOE 2.0, a genetic epidemiologic study of early childhood oral health among a community-based sample of preschool-age children in North Carolina (NC)[54]. In brief, between 2016 and 2019, 8059 children ages 36–71 months who attended public preschools in NC were enrolled in the study and 6404 of them underwent comprehensive clinical examinations by trained and calibrated dental examiners. Data on childhood dental caries experience were collected using modified International Caries Detection and Classification System (ICDAS) criteria[55]. Two supragingival (i.e., "dental plaque") biofilm samples were collected during the clinical encounters immediately prior to dental examinations, which took place before or at least 30 min after snack or breakfast. The plaque sample that was carried forward to microbiome analyses was obtained using sterile toothpicks from the facial/buccal surfaces of primary teeth in the upper-right quadrant, i.e., Universal tooth numbering system: #A, #B, #C, #D, and #E; FDI tooth numbering system: #55, #54, #53, #52, and #51. Upon collection, plaque samples were placed in RNA*later* TissueProtect 1.5 mL tubes and were frozen at −20 °C on-site until transferred to the university core biospecimen processing facility for further processing or long-term storage at −80 °C. Families were provided with a $20 gift card as compensation for their time. Detailed information regarding sample collection, storage, processing, nucleic acid extraction, and sequencing has been reported in a recent protocol publication[56].

### Participants and phenotyping

We carried forward to whole genome shotgun sequencing (WGS, metagenomics/MTG) and RNA sequencing (RNA-seq, metatranscriptomics/MTX) supragingival samples of the first 300 ZOE 2.0 participants, 50% with and 50% without person-level dental caries experience defined at the "established" caries lesion detection threshold, ICDAS ≥ 3)[48]. This lesion threshold corresponds to macroscopic tooth structure loss, i.e., a "cavity". The ZOE 2.0 participants (mean age = 52 months) formed the 'discovery' sample whereas the 'replication' sample comprised 116 similar-aged children (mean age = 55 months), members of the same study population (i.e., enrolled in NC public preschools) examined under virtually identical conditions (i.e., by one clinical examiner) during the parent study's pilot phase. The distribution of male/female participants was balanced in the study, with 208 children in each stratum (Supplementary Table 5). This information was obtained from questionnaires completed by parents or legal guardians which was then cross-checked with biological sex data available via genotyping. For the purposes of the present study, we quantified caries experience using the most sensitive clinical criteria including the enumeration of early-stage caries lesions (i.e., at the ICDAS ≥ 1 threshold) according to the recent international consensus definition of early childhood caries (ECC)[8]. This was done both locally (i.e., within the five tooth surfaces where plaque biofilm was harvested from) and at the person-level (i.e., in the entire dentition, comprising all 88 primary tooth surfaces). We considered the surfaces where plaque was collected from as the most informative in terms of microbiome taxonomy and thus localized caries experience was the primary clinical trait in all analyses. Nevertheless, we posited that the taxonomy of microbiome biofilm in those areas is likely to also be informative for the condition of one's entire dentition—therefore, we considered a secondary 'person-level' caries experience trait as a secondary clinical trait. Estimates of these caries experience traits, in the discovery and replication samples, as well as participants' demographic information are presented in Supplementary Table 5.

## Sequencing and alignment

Total nucleic acid was quantified using QuantIT® PicoGreen®. 5 ng of genomic DNA was processed using the Nextera XT DNA Sample Preparation Kit (Illumina). Target DNA was simultaneously fragmented and tagged using the Nextera Enzyme Mix-containing transposome that fragments the input DNA and adds the bridge PCR (bPCR)-compatible adaptors required for binding and clustering in the flow cell. Next, fragmented and tagged DNA was amplified using a limited-cycle PCR program. In this step index 1(i7) and index 2(i5) were added between the downstream bPCR adapter and the core sequencing library adapter, as well as primer sequences required for cluster formation. The thermal profile for the amplification had an initial extension step at 72 °C for 3 min and initial denaturing step at 95 °C for 30 s, followed by 15 cycles of denaturing of 95 °C for 10 s, annealing at 55 °C for 30 seconds, a 30 second extension at 72 °C, and final extension for 5 min at 72 °C. The DNA library pool was loaded on the Illumina platform reagent cartridge (Illumina) and on the Illumina instrument. Sequencing output from the Illumina HiSeq 4000 2 × 150 was converted to fastq format and demultiplexed using Illumina Bcl2Fastq 2.20.0 (Illumina, Inc. San Diego, CA, USA.). Quality control of the demultiplexed sequencing reads was verified by FastQC (Babraham Institute. Cambridge, UK). Adapters were trimmed using Trim Galore (Babraham Institute. Cambridge, UK). The resulting paired-end reads were classified with Kraken2[57] and Bracken 2.5[58] using a custom database including human, fungal, bacterial, and the expanded Human Oral Microbiome Database (eHOMD) genomes[59] to produce an initial taxonomic composition profile. All reads identified as 'host' were eliminated. Paired-end reads were joined with vsearch 1.10.2[60]. Any remaining adapter reads were trimmed again using Trim Galore. Estimates of gene family, path abundance, and path coverage were produced from the remaining reads using HUMAnN3[61] based on taxonomic estimates from MetaPhlAn 3[62,63].

RNA isolation was performed using Qiagen RNeasy Mini Kit (Cat. No. / ID: 74104) and RNA was quantified using NanoDrop1000 (ThermoFisher, Waltham, MA). To generate MTX data via RNA-seq, sequencing output from the Illumina HiSeq 4000 2 × 150 platform was converted to fastq format and demultiplexed using Illumina Bcl2Fastq 2.20.0 (Illumina, Inc. San Diego, CA, USA). Quality control of the demultiplexed sequencing reads was verified by FastQC (Babraham Institute. Cambridge, UK). Adapters were trimmed using Trim Galore (Babraham Institute. Cambridge, UK). The resulting paired-end reads were classified with Kraken2[57] and Bracken 2.5[58] using a custom database including human, fungal, bacterial, and the expanded Human Oral Microbiome Database (eHOMD) genomes[59] to produce an initial taxonomic composition profile. All reads identified as host were eliminated. Paired-end reads were joined with vsearch 1.10.2[60]. The resulting single-end reads were again trimmed of any remaining adapters using Trim Galore. HUMAnN 3.0[61] was used to generate gene family and pathway-level data based on taxonomic estimates from the MetaPhlAn 3[62,63] metagenomic analysis. Additionally, MTX gene expression analyses were performed for the four "top species" that were tested experimentally. Reads classified by Kraken2 as belonging to each of the four top species and any strain thereof were extracted by species, individually aligned to each species' relevant reference transcriptome with STAR[64] (v.2.7.10b), and subsequently quantified via Salmon[65] (v.1.10.1).

## Quality control procedures

An overview of quality control, scaling, and filtering of taxa is presented in Supplementary Fig. 7. After the removal of viral sequences, the remaining data were first arranged in a reads-per-kilobase (RPK) format, then rescaled into transcripts-per-million (TPM), and finally rescaled close to the averaged-over-subject RPK level. The average RPK in MTG in the discovery sample was 8,004,958, and the rescaled TPM was set as 8,000,000 for each subject. To facilitate discovery and inference, species with relative abundance less than $10^{-5}$ or with a prevalence rate less than 10% were excluded from all taxonomic discovery analyses. Out of 6411 non-viral species, 5990 low-abundance and 2935 low-prevalence (including 2932 both low-abundant and low-prevalence) taxa were filtered, retaining 418 taxa for all downstream MTG analyses. MTX data in the discovery sample, as well as MTG and MTX data in the replication sample were all processed in a similar fashion. Total RPKs per subject were 11,014,832, 5,077,759, and 2,739,606 on average, and the total rescaled TPMs per subject were on average 11 million, 5 million, and 3 million, respectively. The numbers of retained taxa after applying abundance and prevalence filters were 385 for the MTX in the discovery sample, 422 and 397 for the MTG and MTX data in the replication sample, respectively. We refer to these sets of retained taxa as "core." TPM normalization was not done for the targeted MTX of the 4 top species, due to limited diversity of the sample. Average RPK per sample was 72,474. We retained genes with average >20 reads for downstream differential expression and gene-gene interaction analyses. Using this filter, out of 9103 total genes available for these 4 top species, 542 genes were retained: 47 for *S. mutans*, 39 for *S. sputigena*, 8 for *P. salivae*, and 448 for *L. wadei*.

## Identification of species significantly associated with caries experience

We used log-normal linear models to test the association between species' differential abundance in MTG and MTX data and quantitative measures of caries experience. The models included terms for phenotypes of interest (i.e., either localized or person-level caries quantitative experience), batch effects (i.e., the first sequencing batch included 52 samples and the second batch the remaining 248 samples), age at enrollment (measured in months), and race/ethnicity (reported by legal guardians and categorized as white-non Hispanic, African American-non-Hispanic, and others including Hispanics), and unity was added to the rescaled TPM abundance data before log-transformation with base 2. The presence of each species in MTG data was controlled for in models examining differential expression (i.e., abundance in MTX data). A false discovery rate (FDR) correction for multiple testing was applied using the Benjamini-Hochberg procedure[66] for each of the four models (caries traits and abundance/expression).

We used strict criteria for the identification of bacterial species associated with caries experience and required taxa to be FDR-significantly differentially abundant for quantitative disease experience defined both locally and at the person level, in MTG and MTX data (i.e., 4 models), in the discovery sample ($n = 300$). To reduce the potential for false discovery, we sought for additional evidence of association in an independent sample, i.e., we examined the replication of the previously identified associations in a sample of 116 similarly aged participants from the same population as the parent study (i.e., public preschools in NC), similar clinical and microbiome data. Thus 4 more models were created, for localized and person-level disease, in MTG and MTX data. As evidence of replication, we considered, in order of ascending importance, directional consistency of the estimate of association, nominal significance, or FDR-level significance in the replication sample. Species that were FDR-significant in all 4 models in the discovery sample and were at least nominally significant for localized disease experience in MTG data were termed "significant species". This set of species with high-confidence evidence of association from multiple traits, MTG, and MTX data, and from all 416 study participants were prioritized for reporting and were candidates for consideration in the experimental validation pipeline. The raw data emanating from microbiome analyses that were used to create Figs. 2 and 3, Supplemental Figs. 1 and 2, and Supplemental Tables 1, 2, 3, and 4, as well model coefficients' 95% confidence intervals (upper and lower bounds, where applicable) and variable legends, are included in the Supplemental Dataset.

## Inter-species correlations and pathways involving species significantly associated with caries experience

To provide initial insights into the inter-relationships of the 16 significant species, we examined their pairwise correlation patterns in health and disease. For this purpose, we used Pearson correlation coefficients between model residuals that were generated for each of the 16 significant species. These log-normal models had the same specifications as models used in the main analyses (i.e., adjusted for batch effects and age) with the addition of an adjustment for disease experience. Species' mean correlations with each of the other 15 significant species were examined between health (i.e., no localized disease experience) and disease (i.e., any localized disease experience), and in MTG and MTX.

To identify pathways significantly associated with caries experience in the biofilm transcriptome, we examined pathway and pathway-species MTX data that were prepared in RPK format as described above. Total RPKs per sample were on average 383,119, and the RPK data were transformed to TPM format with a scale of 400,000, as previously described. Each pathway, summed over all species including those unclassified, was tested using the log-normal model with the same set of covariates as in the main discovery analysis and the addition of a unity before $\log_2$ transformation. An FDR correction for testing 297 pathways was applied using the Benjamini-Hochberg procedure. Using this procedure, we identified pathways whose expression in MTX was significantly associated with caries experience. Additionally, we examined the representation of the 16 significant species in these pathways as percent of significant over all species involved in each pathway.

## Shortlisting of species for virulence assays and experimental testing

We departed from a list of 16 species that showed strong and replicable evidence of association with dental caries experience (Supplementary Table 1). To select a shortlist of candidates that could be fully characterized in virulence, biofilm, and possibly in vivo studies in this study, we considered species' statistical evidence of association in the discovery sample (i.e., p-value), evidence replication in the independent sample, representation of different genera (i.e., selected one species per genus), and availability of clinical isolates. Based on these criteria, from the list of 16 significant species we prioritized *Streptococcus mutans* (the known, well-established pathogen), and *Selenomonas sputigena*, *Prevotella salivae*, and *Leptotrichia wadei* (as 3 new candidates). These 4 taxa, referred to as "top species", were carried forward to in vitro virulence assessments and biofilm characterization and served as candidates for further in vivo colonization and cariogenicity studies.

## Microorganisms and growth conditions used in laboratory validation

*Streptococcus mutans* UA159 (ATCC 700610), *Selenomonas sputigena* (ATCC 35185), *Leptotrichia wadei* (JCM 16777), *Prevotella salivae* (JCM 12084) were used in in vitro and in vivo studies. Bacteria were grown in Brain Heart Infusion Supplemented (BHIS) medium (Brain Heart Infusion broth supplemented with 5 g/L yeast extract, 5 mg/L Hemin, 1 mg/L Vitamin K1, and 0.5 g/L L-cysteine) at 37 °C to exponential phase in an anaerobic chamber (Anaerobe Systems). For in vitro biofilm studies, a saliva-coated hydroxyapatite disc ($2.7 \pm 0.2$ cm²; Clarkson Chromatography Products) was placed in a vertical position to mimic the tooth-enamel surfaces of human teeth. Single or mixed biofilms were grown on the apatitic surfaces in BHIS medium ($10^7$ CFU/mL for the new species and/or $10^5$ CFU/mL *S. mutans*) supplemented with 1% sucrose, the primary dietary sugar associated with tooth decay, to simulate cariogenic conditions relevant to children with ECC. Biofilms were grown on saliva-coated hydroxyapatite surfaces for up to 24 h to allow the establishment of single or mixed-species communities. For growth

dynamics experiments, biofilms were examined at 4 h (i.e., initial attachment stage) and at 10 h (i.e., intermediate stage). For real-time live imaging, a green fluorescent protein (GFP)-tagged *S. mutans* UA159 strain (*S. mutans* UA159 Ef-Tu-gfp)[67] was used. Biofilm EPS glucan matrices were labeled via supplementing the culture medium with 1 μM Alexa Fluor 647 dextran conjugate (Molecular Probes) during biofilm growth. This labeling method is highly specific for *S. mutans*-derived α-glucans since the fluorescently labeled dextrans serve as primers for *streptococcal* glycosyltransferases and are directly incorporated into glucans during biofilm EPS synthesis. *S. sputigena* single-species biofilms used in the motility tracking assays were grown for 24 h in BHIS (1% sucrose) supplemented with cell-free purified GtfB enzymes (20 U).

## Top species' virulence and metabolic profile assessment

**Acid tolerance and acidogenicity.** Acid tolerance tests were performed using Brain Heart Infusion (BHI) broth (Anaerobe Systems) with pre-adjusted pH. Lactic acid (13.42 M) was used to adjust the pH of the medium (ranging between 2.9 and 6.5). Each of the top species was serially diluted to $10^7$ CFU/mL. One hundred microliters (100 μL) of bacterial suspension were transferred to 900 μL pH adjusted medium to study the growth of single species under acidic conditions. In parallel, each of the new species (*S. sputigena*, *L. wadei*, or *P. salivae*) was mixed with *S. mutans* (equal proportion of single species) to examine the growth of mixed species combinations. The cultures were incubated in the anaerobic chamber at 37 °C for 48 h. Bacterial growth was quantified using a microplate reader (SpectraMax M2e) by measuring optical density values at 610 nm, whereas the pH values were measured using a pH meter (FiveEasy Benchtop F20 pH/mV Meter, Mettler Toledo). Both the lowest pH values at which each bacterial species survived (detectable growth) and the final pH values of the culture (after 48 h) were recorded to assess acid tolerance. For acidogenicity (acid production), we used standard glycolytic pH-drop assay. Bacterial cells (a final number of viable cells of $10^7$ CFU/mL, single or mixed-species, with equal numbers, as described above) were incubated in salt solution (50 mM KCl, 1 mM $MgCl_2 \cdot 6H_2O$) with 1% glucose and allowed to produce acids over time. The decrease in pH was measured using a pH meter every 1 h over a 12 h period during which each species actively produced acids. The average proton production rate over the 12 h period was calculated to compare acidogenicity of different species either alone or in combination with *S. mutans*.

**Metabolic profiling.** Top species' metabolic profiles were measured using real-time isothermal microcalorimetry[68]. *S. mutans*, *S. sputigena*, *L. wadei*, *P. salivae*, and *P. oulorum* were transferred to BHI broth and incubated in the anaerobic chamber at 37 °C for 48 h. Each culture was serially diluted to $10^7$ CFU/mL, and 60 μl of each bacterial suspension was transferred, in triplicate, in titanium vials (calWell; SymCel) containing 540 μl of pre-reduced BHI. Real-time heat production, proxying metabolic activity, was measured using a calScreener™ microcalorimeter (SymCel Sverige AB, Stockholm, Sweden) in a 48-well plate (calPlate™) continuously for up to 48 h, as previously described[69]. The full processing of the samples and plate preparation were performed inside the anaerobic chamber, to maintain the anaerobic condition for optimal bacterial growth. Heat and corresponding energy data were quantified with calView™ software (Version 1.0.33.0, © 2015 SymCel, Sverige AB). The instrument was set and calibrated at 37 °C, with all handling and set-up done according to the manufacturer's recommendations.

**Biofilm live imaging.** Biofilm live imaging was performed based on our established fluorescence labeling and confocal imaging protocols optimized for oral biofilms with some modifications[70]. Biofilms were dip-washed three times in phosphate buffered saline (PBS, pH 7.1) to remove any loosely bound microbes from the surface. To enhance the

GFP fluorophore development in GFP-tagged *S. mutans* cells, we performed aerobic fluorescence recovery immediately before imaging, following the protocol previously reported[71]. Biofilms were counterstained with SYTO82 (Molecular Probes), which labeled all bacterial cells. Super-resolution live imaging was conducted at 37 °C using a 40× (numerical aperture = 1.2) water immersion objective on a Zeiss LSM800 upright confocal microscope with Airyscan. For real-time imaging of bacterial motility, multi-channel confocal images were taken at a 2.6-second interval for 30 s.

**Computational biofilm image analysis.** We then used a fluorescence subtraction method to analyze the biofilm spatial structuring (positioning of different species and EPS across the 3D biofilm structure) as detailed previously[38]. In brief, we applied channel subtraction using the Image Calculator in ImageJ Fiji v.2.11.0 (https://imagej.net/Fiji) to calculate the fluorescence signal from the new species in the mixed-species biofilm, using the following equation: $Ch_{New\ species} = Ch_{All\ bacteria} - Ch_{S.\ mutans}$, where $Ch_{All\ bacteria}$ is the channel of all bacteria (SYTO82), and $Ch_{S.\ mutans}$ is the channel of *S. mutans* (GFP). Computational image processing and quantitative analysis were performed using BiofilmQ software v.0.2.2 (https://drescherlab.org/data/biofilmQ), an image analysis toolbox optimized for biofilms[29]. After image segmentation using Otsu algorithm, we conducted a cube-based object declumping in BiofilmQ that dissected the entire biofilm into small cubic volumes (cube size = 2 μm). This function allows analyses of structural properties within biofilm subdomains with spatial resolution since each cube has a unique spatial coordinate in 3D. Parameters including local shape volume, relative abundance, and intensity were calculated for each channel within the cubes. For co-localization analysis, we used Mander's overlap coefficient in BiofilmQ to quantify the degree of spatial proximity of two different fluorescence signals in relation to each other. Two Manders' coefficients were calculated: 1) the new species with *S. mutans* and 2) the new species with EPS. To track single-cell bacterial motility in real time, we performed computational single-particle tracking and generated time-resolved trajectories using the TrackMate plugin in ImageJ Fiji[72]. Biofilm visualization was performed using maximum intensity projection and 3D surface rendering in ImageJ Fiji.

**In situ biofilm pH measurement and visualization.** We used a pH-responsive fluorescent dye (C-SNARF-4) combined with confocal live imaging to determine the in situ pH at the biofilm-apatite interface as described previously[39]. The C-SNARF-4 exhibits a pH-dependent emission shift, allowing measurement of localized biofilm pH using the ratio of the fluorescence intensities at two emission wavelengths. Briefly, 24 h biofilms were dip-washed and equilibrated in HEPES buffer (50 μM; pH = 7.4) for 60 min. Then, biofilms were stained with 30 μM C-SNARF-4 followed by continuous ratiometric imaging at two detection wavelength ranges (576–608 nm and 629–650 nm). The first image (t = 0 min) was taken before glucose exposure and image acquisition was performed every 5 min at 37 °C after adding glucose (1% final concentration), using a 20x (numerical aperture = 1.0) water dipping objective on the Zeiss LSM800 confocal microscope. Calibration of the ratiometric probe was conducted using HEPES buffer (50 μM; pH = 4.5 to 7.5 in steps of 0.5 pH unit). Computational image analysis and visualization were performed using ImageJ Fiji.

**Ex vivo human tooth-enamel biofilm model and enamel surface analyses.** We employed an ex vivo human tooth-enamel model which allows analysis of the extent of enamel decay underneath the biofilm[40]. Briefly, single or mixed-species biofilms (67 h) were formed on sterilized enamel specimens (4 × 4 mm) prepared from de-identified extracted human teeth. The biofilm was removed using enzymatic treatment (dextranase and mutanase) followed by water bath sonication, which was optimized for biofilm removal without causing artificial surface damage[40]. The macroscopic demineralized areas on the

enamel surface (similar to those found clinically manifest early childhood caries) were visualized using stereomicroscopy (ZEISS Axio Zoom V16). Then, the surface topography and roughness of the tooth enamel surface were assessed by non-destructive confocal microscopy using a 50x (numerical aperture = 0.95) objective on the Zeiss LSM800 microscope and ConfoMap software (version 7.3.7984). Next, the human enamel specimens were mounted on acrylic rods and sectioned (100 ± 20 μm thickness) with a hard tissue microtome (Silverstone-Taylor Hard Tissue Microtome, Series 1000 Deluxe) for transverse microradiography. The sections were placed in the TMR-D system and x-rayed (45 kV; 45 mA) at a fixed distance for 12 s. An aluminum step wedge was x-rayed under identical conditions. The digital images were analyzed using TMR software v.3.0.0.18, with sound enamel defined at 87% mineral volume[73].

**Targeted bacterial gene expression and gene-gene interaction analyses.** We used a targeted MTX approach to test *S. mutans*, *S. sputigena*, *L. wadei*, and *P. salivae* gene expression. Reads were aligned to these species reference genomes using STAR-Salmon[64,65]. Where multiple strains were available, results were merged to the species level. The association of gene expression with caries experience was then tested for each gene using log-normal models adjusting for batch effects, age, and race/ethnicity, and applying an FDR multiple testing correction for each species. Additionally, we examined gene-gene interactions between *S. mutans* and *S. sputigena*—the two species that demonstrated enhanced acidogenesis and unique biofilm structure when co-cultured. To this end, the same log-normal models including gene-gene interactions were used including pairwise combinations of all filtered genes from these two species, applying an FDR multiple testing correction.

**In vivo rodent model of childhood caries.** Bacterial colonization on teeth and their impacts on disease onset were assessed on an established rodent caries model as detailed elsewhere with some modifications[43,74]. In brief, 15-d old female Sprague-Dawley rat pups (specific-pathogen-free grade) were purchased with dams (8 pups per dam) from Harlan Laboratories (Indianapolis, IN, USA). Upon arrival, animals were pre-screened for oral infection of *S. mutans* and *S. sputigena* by oral swabs and real-time polymerase chain reaction (qPCR) and were determined not to be infected with either organism. Oral swabs (FLOQSwab, COPAN Diagnostics, Murrieta, CA, USA) were taken from each of the animals and bacterial DNA were extracted using DNeasy PowerLyzer Microbial Kit (Qiagen, Valencia, CA, USA). Species-specific qPCR primer sets were used for microorganism detection as follows: *S. mutans*: Forward 5′ ACCAGAAAGGGACGGCTAAC 3′, Reverse 5′ TAGCCTTTTACTCCAGACTTTCCTG 3′; *S. sputigena*: Forward 5′ GGTCAGCCTTATCAGTTCCGTT 3′, Reverse 5′ GGCGAGCTTTC AGCAATCTTAG 3′; all bacteria: Forward 5′ TCCTACGGGAGGCA GCAGT 3′, Reverse 5′ GGACTACCAGGGTATCTAATCCTGTT 3′.

The pups were randomly assigned into four groups that received different bacterial infections and were housed separately: (1) *S. mutans* alone; (2) *S. sputigena* alone; (3) *S. mutans* plus *S. sputigena;* (4) control without *S. mutans* or *S. sputigena* infection. Animals were inoculated daily using cotton oral swabs with actively growing cultures of *S. mutans* and/or *S. sputigena* (~10^8 CFU/mL in BHIS) at the age of 19–23 d (five doses in total). Each of the infected groups were confirmed for its designated microbial infection at 21 d, 24 d, and 30 d by oral swabs and qPCR, while the control group remained free of either *S. mutans* or *S. sputigena*. No cross-contamination was detected throughout the experiment. All animals were provided with the National Institutes of Health cariogenic diet #2000 and 5% sucrose water *ad libitum* to mimic the cavity-promoting diet relevant to childhood caries. The experiment proceeded for three weeks, at the end of which the animals were euthanized using carbon dioxide. The jaws were dissected and processed for caries scoring of teeth according to Larson's modification of Keyes' system[75]. Caries scores were determined by a calibrated

examiner. Investigators were masked to experimental group (i.e., infection) allocations during the infection, sampling, and assessment stages, by using color-coded samples. In vitro and in vivo experimental data were presented as mean ± standard deviation. Data were subjected to Student's pairwise $t$-test or analysis of variance (ANOVA) with post-hoc Tukey HSD test to account for multiple comparisons. Differences between groups were considered statistically significant when $p < 0.05$.

## Reporting summary
Further information on research design is available in the Nature Portfolio Reporting Summary linked to this article.

## Data availability
Raw DNA and RNA sequence data for ZOE 2.0 have been deposited and are publicly available as part of dbGaP accession phs002232.v1.p1 "TOPDECC-Trans-omics for Precision Dentistry and Early Childhood Caries: Genome-Wide Genotyping (CIDR) and Microbiome in the ZOE 2.0 Study" and the Sequence Read Archive (SRA) as BioProject 671299 (PRJNA671299; dbGaP: phs002232; https://www.ncbi.nlm.nih.gov/bioproject/671299). Raw DNA and RNA sequence data for ZOE pilot (the replication sample) have been deposited and are publicly available as part of BioProject 843091 "ZOE 2.0 pilot study" (PRJNA843091). Microbiome taxonomy (MTG and MTX) and pathway information for the entire biofilm microbial community, as well as targeted MTX data for the four top species used in this study have been deposited and are publicly available alongside metadata (i.e., demographic and clinical phenotype information) via the Carolina Digital Repository and accession number 5d86p890x. Reference genomes were obtained from the expanded Human Oral Microbiome Database (https://www.homd.org/). The source data file used to generate figures and the tables is available in the Supplementary Information. Source data are provided with this paper.

## Code availability
The scripts used to perform this analysis can be found at https://doi.org/10.5281/zenodo.7707297[76].

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

## Acknowledgements

This work was supported by research grants from the National Institutes of Health: National Center for Advancing Translational Sciences National UL1TR001111 (K.D.); National Institute for Dental and Craniofacial Research U01DE025046 (K.D.), R01DE025220 (H.K.), and R03DE028983 (D.W.); Z. R. is supported by the NIDCR Postdoctoral Training Program under award number R90DE031532. Z. R. was a recipient of the Colgate–Palmolive Fellowship. The UNC Microbiome Core is funded in part by the Center for Gastrointestinal Biology and Disease (CGIBD P30 DK034987) and the UNC Nutrition Obesity Research Center (NORC P30 DK056350). The content is solely the responsibility of the authors and does not necessarily represent the official views of the funders. The authors thank Dr. Patricia V. Basta and her team at the UNC-Chapel Hill Biospecimen Processing facility for the accessioning, storage, and disbursement of the supragingival biofilm microbiome specimens in the ZOE studies; Dr. Anderson Hara at Indiana University for technical assistance in transverse microradiography; Dr. Justin Merritt at Oregon Health & Science University for kindly providing the *S. mutans* UA159 Ef-Tu-gfp strain; and Dr. Lamprini Karygianni and Dr. Thomas Thurnheer at University of Zurich for the gift of *S. sputigena* ATCC 35185 strain. Some schematic diagrams in the figures are created with BioRender.com (Publication License #RS251UWR8N).

## Author contributions

K.D., K.E.N., A.A.R., D.W., and H.K. designed the study. K.D., M.A.S.P, P.S., J.G., and A.G.F.G participated in data collection and sampling. J.R. and M.A.A.P. carried out sample processing and sequencing. H.C., J.R., B.M.L, C.L., A.O., and D.W. conducted bioinformatics analyses. Z.R., A.A.R., and H.K. carried out in vitro experiments. Z.R. and H.K. carried out in vivo experiments. H.C., Z.R., K.D., A.A.R., D.W., and H.K. wrote the paper with contributions from all the authors.

## Competing interests

The authors declare no competing interests.

## Ethical approval

Human observational data and analyses received approval (#14-1992) from the University of North Carolina-Chapel Hill Office of Human Research Ethics Institutional Review Board on September 18, 2014. Legal guardians of all children provided written informed consent for participation in the study. The in vivo experimental study was reviewed and approved by the Institutional Animal Care and Use Committee of the University of Pennsylvania (IACUC#805735). All research was performed in accordance with the Declaration of Helsinki.

## Additional information

[1]Department of Biostatistics, Gillings School of Global Public Health, University of North Carolina at Chapel Hill, Chapel Hill, NC, USA. [2]Biofilm Research Laboratories, Center for Innovation & Precision Dentistry, School of Dental Medicine, University of Pennsylvania, Philadelphia, PA, USA. [3]Division of Pediatric and Public Health, Adams School of Dentistry, University of North Carolina at Chapel Hill, Chapel Hill, NC, USA. [4]Department of Epidemiology, Gillings School of Public Health, University of North Carolina at Chapel Hill, Chapel Hill, NC, USA. [5]UNC Information Technology Services and Research Computing, University of North Carolina at Chapel Hill, Chapel Hill, NC, USA. [6]UNC Microbiome Core, Center for Gastrointestinal Biology and Disease, School of Medicine, University of North Carolina at Chapel Hill, Chapel Hill, NC, USA. [7]Department of Medicine, Division of Gastroenterology and Hepatology, School of Medicine, University of North Carolina at Chapel Hill, Chapel Hill, NC, USA. [8]Artificial Intelligence Innovation Lab, Cedars-Sinai Medical Center, Los Angeles, CA, USA. [9]Department of Comprehensive Care, School of Dental Medicine, Tufts University, Boston, MA, USA. [10]Division of Diagnostic Sciences, Adams School of Dentistry, University of North Carolina at Chapel Hill, Chapel Hill, NC, USA. [11]Division of Oral and Craniofacial Health Sciences, Adams School of Dentistry, University of North Carolina at Chapel Hill, Chapel Hill, NC, USA. [12]Department of Orthodontics, School of Dental Medicine, University of Pennsylvania, Philadelphia, PA, USA. [13]These authors contributed equally: Hunyong Cho, Zhi Ren. ✉e-mail: Kimon_Divaris@unc.edu; did@email.unc.edu; koohy@upenn.edu

