## [Peer Review File · Nature Communications]

REVIEWER COMMENTS

Reviewer #1 (Remarks to the Author):

In this manuscript, the authors study the taxa associated with carries present in pediatric clinical samples. Their analysis identified three unique species that may influence pathogenesis. They studied these species in vitro and found that co-culture can influence metabolism and biofilm structural dynamics. Finally, in a murine infection model they illustrate the importance of the *S. mutans*, *S. sputigena* co-culture for disease. This was a very interesting, data-rich manuscript. Strengths include the initial metagenomics/transcriptomics analysis and the murine infection studies. The major weakness was key missing in vitro data and an overinterpretation of some of the biofilm co-culture results.

1. p.4 line 104- FDR needs to be defined.
2. The data presented in Figure 4 A1 and A2 appear to not represent actual proton production rates. The value is calculated after 48 h of culture incubation. Each of the mono- and co-cultures will probably enter stationary phase at different time points. Additionally
3. Key missing data for figure 4 A1 and A2 are actual final cell #s for each of the species in mono- and co-culture. What is the yield of cells? Are both partners actively growing in the co-culture? These are key data bot for interpreting what's presented in the rest of the Figure as well as experiments that occur later on.
4. The data presents in Figure 4C1 indicate that the measurements made in A12 and B12 occurred at 48 h before growth was finished for *Ss* and *Ps*.
5. The figures in general are hard to read. This takes away from the study in my opinion. In many places the text is way too small and the complexity of the figures are a bit overwhelming. For example, in Figure 3 the heat map scale is not defined.
6. P.5 lines 120-129. This section needs to be better developed. Of the pathways implicated by MTX how many of the 16 significant species possess them? A brief explanation of what each pathway is involved in would help the reader too (i.e. UDP GlcNac biosynthesis is important for peptidoglycan synthesis and cell growth).
7. Regarding the data in Fig. 5- Presumably the co-culture biofilm reactors were inoculated with the two species being present in the original inoculum. The single species data from this same figure also suggests an initial attachment deficiency for the three new non-mutans species. Is this true? If so differences in attachment efficiencies probably are really important for dictating later biofilm structure. What does the initial attachment stage look like for mono- and co-cultures?

8. Nowhere in the materials and methods, text, or figure legends does it indicate how long the biofilms have been grown that are displayed in Figure 5??
9. A time course of biofilm development should be shown for the Sm, Ss, and Ss + Sm biofilms.
10. The data suggesting *S. sputigena* becomes trapped in the glucan matrix produced by *S. mutans* are not convincing or definitive. The best data related to this point are presented in Figure 6EF. These data are only suggestive and the conclusions made by the author are speculative.

Reviewer #2 (Remarks to the Author):

This work addressed a global oral health problem that compromises quality of life, namely childhood dental caries. The etiological agent underlying dental caries is the microbial biofilm developing on the tooth surface, particularly when this micro-environment acquires an acidophilic and acidogenic profile. Mutans streptococci are long known to be strongly associated with the disease, yet the list of the involved novel taxa is increasing, thanks to the application of high-throughput nucleotide or aminoacid sequencing techniques. This particular study used human dental biofilm samples from preschool-age children as well as an experimental rodent model. The metagenomics analysis identified 16 taxa strongly associated with childhood caries. Perhaps the main highlight of the study is the identification of the previously unrecognized ability of *S. sputigena* to exacerbate disease severity in vivo when co-infected with *S. mutans*.

The study is very well designed with multiple levels of experimentations and approaches, and is expected to have an important contribution to the literature. I would further recommend the following:

1. Other newly discovered important taxa have also been shown cooperate with *S. mutans*. Discuss finding of other studies whereby *Scardovia wiggsiae* in particular is shown to synergise with *S. mutans* (PMID: 28930642, PMID: 28930642)
2. Discuss findings of other studies whereby other commensal taxa, such as *S. oralis*, are shown to antagonize with *S. mutans* (PMID: 29327482)
3. The title is rather generic and does not directly acknowledge the species or the disease. Therefore, "*S. sputigena*" should be added next to "pathobiont" and "oral disease" should be replaced "caries".

Reviewer #3 (Remarks to the Author):

By employing a new multimodal pipeline, in this study, the authors explored novel disease-relevant inter-species interactions and pathogenic mechanisms in preschool-age children. The authors first identified 16 taxa strongly associated with childhood caries. Among them, the authors found a novel inter-species interaction and unique spatial structuring in which a motile flagellated species (i.e., *S. sputigena*) becomes immobilized in the EPS matrix produced by disease-causing pathobionts (i.e., *S. mutans*) and proliferates to build a 3D multicellular superstructure with enhanced acidogenesis. The authors further demonstrated that the interaction between *S. sputigena* (SS) and *S. mutans* (SM) augments the severity of dental caries *in vivo*, suggesting a new pathobiont exacerbating biofilm virulence.

I enjoyed reading this manuscript. The quality of the writing and the presentation of the figures was superb and I really appreciate the authors efforts and attention to detail. Overall, I felt the overall message is clear and the work is impactful in the field, but needs to be improved with some critical experiments and logical controls to support their major conclusion. I have outlined the major and other specific concerns below:

Major concerns:

1. In Figure 4, by co-culturing SS and SM, the authors showed a synergistic effect in the acid production by them. Does the SS activate the acid production from SM? or the other way around? , or both? What are the molecular mechanisms behind this metabolic activation? As the authors claimed in the text, if pathobiont-mediated spatial structuring enhances biofilm virulence, the authors should provide the data demonstrating that special structuring is required for enhancement of the virulent functions such as acid generation.

2. Related to the previous comment, does the Δ gtfC mutant of SM fail to enhance the biofilm virulence? How about the deletion of SS's motility-related gene flagellin in the context of the enhancement of biofilm virulence? To prove the usefulness of their experimental design shown in Figure 1, the authors should validate the functional impact of those genes (at least one related gene) *in vitro* and *in vivo*, and identify the mechanistic link. Just providing the association is not sufficient.

Other concerns,

3. The text and content of Figure 1 do not match completely. As a result, the readers may miss where the authors describe in the manuscript. The authors should resolve this issue.

4. In Figure 8, to strengthen the specificity of SS, the author should provide negative controls such as *P. salivae* (PS) that do not increase the proton production rate when co-culturing with SM.

Response to Reviewers Comments:

Reviewer #1:

1.1-In this manuscript, the authors study the taxa associated with carries present in pediatric clinical samples. Their analysis identified three unique species that may influence pathogenesis. They studied these species in vitro and found that co-culture can influence metabolism and biofilm structural dynamics. Finally, in a murine infection model they illustrate the importance of the *S. mutans*, *S. sputigena* co-culture for disease. This was a very interesting, data-rich manuscript. Strengths include the initial metagenomics/transcriptomics analysis and the murine infection studies. The major weakness was key missing in vitro data and an overinterpretation of some of the biofilm co-culture results.

Authors' response:

We thank the reviewer's comments and suggestions for improvement. We have addressed each of them and made revisions in the manuscript accordingly.

1.2-p.4 line 104- FDR needs to be defined.

Authors' response:

We have now defined FDR in the manuscript, in the beginning of the results section as follows: "We used strict, multiple testing (i.e., false discovery rate, FDR)-controlled, across-trait criteria to identify this set of 16 disease-associated species."

1.3-The data presented in Figure 4 A1 and A2 appear to not represent actual proton production rates. The value is calculated after 48 h of culture incubation. Each of the mono- and co-cultures will probably enter stationary phase at different time points.

Authors' response: We thank the reviewer for raising this important point and for providing an opportunity to clarify it. The proton production rates were in fact calculated using a standard glycolytic pH drop assay and not from 48 h culture incubation. The dynamic pH changes were determined over 12h (example of the obtained pH curves is shown below) using bacterial cells at the mid-exponential phase (to ensure that each species was actively growing and producing acids), rather than using a single measurement or end-point measurement. We realize this point was unclear in the original manuscript and we have now clarified it. On the other hand, we do agree with the reviewer that the actual proton production rate is not constant over time. To address these points ensemble, we provided an additional description in the manuscript for clarification and now refer in the text to "average proton production rate" instead of "proton production rate".

1.4- Additionally, Key missing data for figure 4 A1 and A2 are actual final cell #s for each of the species in mono- and co-culture. What is the yield of cells? Are both partners actively growing in the co-culture? These are key data but for interpreting what's presented in the rest of the Figure as well as experiments that occur later on.

Authors' response:

We apologize for the missing information in the original submission and have now included them in the Materials and Methods section of the revised manuscript. The glycolytic pH drop assay was performed by incubating a final number of viable cells of 10^7 CFU/mL (single or mixed-species with equal numbers) in a salt solution. The salt solution does not increase microbial growth but maintains cell viability and allows each of the species to actively produce acids over time (see pH curves for each species above).

1.5- The data presents in Figure 4C1 indicate that the measurements made in A12 and B12 occurred at 48 h before growth was finished for Ss and Ps.

Authors' response: Indeed, the reviewer is correct to point out that the measurements in Figure 4 A1/A2 and B1/B2 were made between 12h and 48h, before the growth of *S. sputigena* and *P. salivae* was finished (up to 72h, as shown in Figure 4C1). This is an inherent but arguably inconsequential limitation as *Selenomonas* and *Prevotella* are known to be slow growers. The heat flow curves reflect the energy released at a constant temperature during the metabolic processes while energy released is not exclusively a reflection of bacterial growth, but of all metabolic processes. Since *Selenomonas*, *Leptotrichia* and *Prevotella* are strict anaerobes, we were not able to measure their growth curves in the spectrophotometer.

1.6- The figures in general are hard to read. This takes away from the study in my opinion. In many places the text is way too small and the complexity of the figures are a bit overwhelming. For example, in Figure 3 the heat map scale is not defined.

Authors' response: We have improved the readability of the figures by increasing font sizes and clearer labels. In Figure 3, we added the scale legend but did not display the unit as Pearson correlation is a standardized measure. Figure 7 was split into two separate figures to improve readability.

1.7-P.5 lines 120-129. This section needs to be better developed. Of the pathways implicated by MTX how many of the 16 significant species possess them? A brief explanation of what each pathway is involved in would help the reader too (i.e. UDP GlcNac biosynthesis is important for peptidoglycan synthesis and cell growth).

Our response:

We must clarify that initially, we used MTX data to help characterize functions that the top 16 species were involved in and did not carry out a de novo MTX-based pathway discovery analysis for caries experience. For this reason, we had prioritized for testing only pathways with some representation of significant species. Four of these pathways involving significant species (first four rows of Extended Data Table 2) namely PWY-1042, PWY-5100, UDPNAGSYN-PWY, and PWY-6163 were significantly associated with caries experience. To address the reviewer's question, we now examined all pathways' (n=297) associations with caries experience irrespective of the involvement of significant species or abundance. We found that 6 (i.e., 2 additional) pathways were significantly associated with localized quantitative caries experience. Four of these pathways (mentioned above) involved significant species and two (PWY-7234 "inosine-5'-phosphate biosynthesis III" and PWY-6263 "superpathway of menaquinol-8 biosynthesis II") did not. We have updated our Extended Data Table 2 and the narrative text accordingly to include the finding that 4 out of 6 significant pathways involved one or more significant species. Additionally, without expanding this section beyond the scope of helping annotate the identified species, we now provide brief descriptions of two additional pathways (pyruvate fermentation to acetate and lactate II and UDP-N-acetyl-D-glucosamine biosynthesis I) as examples of likely relevance of these functions biological processes known to be present in dental disease.

1.8-Regarding the data in Fig. 5- Presumably the co-culture biofilm reactors were inoculated with the two species being present in the original inoculum. The single species data from this same figure also suggests an initial attachment deficiency for the three new non-mutans species. Is this true? If so differences in attachment efficiencies probably are really important for dictating later biofilm structure. What does the initial attachment stage look like for mono- and co-cultures?

Authors' response: We appreciate the reviewer's insightful comment that helped us clarify some mechanistic aspects underlying our observation. To answer this question, we conducted additional experiments to assess the initial attachment stage (4 hours) for the mono- and co-culture (see figure below) on saliva-coated apatite surfaces. The data demonstrate that *S. sputigena*, *L. wadei*, *P. salivae*, and *S. mutans* bind similarly to the surface at 4 hours. The initial attachment stage of co-cultures (*S. mutans* mixed with each of the new species) also showed similar binding patterns. Based on these findings, the post-attachment co-metabolism and interaction with *S. mutans* appears to be more important for the distinct biofilm spatial structuring observed at 24 hours. We have added these findings in the Results section of the revised manuscript (part of Extended Data Fig. 5).

Figure. Initial attachment stage (4 hours) of single and mixed biofilms (*S. mutans* co-cultured with each new species) on saliva-coated hydroxyapatite.

1.9- Nowhere in the materials and methods, text, or figure legends does it indicate how long the biofilms have been grown that are displayed in Figure 5??

Authors' response: This was an oversight on our end and thank you for providing us with an opportunity to correct it. The biofilms in Figure 5 are grown for 24 hours. We have revised the manuscript to include this information in the Materials and Methods and the Results sections, as well as in the figure legend.

1.10- A time course of biofilm development should be shown for the Sm, Ss, and Ss + Sm biofilms.

Authors' response: As suggested by the reviewer, we have conducted additional time course experiments for biofilm development by each of the 4 species individually or in combination with *S. mutans*. We assessed the initial attachment stage (4 hours) and an intermediate stage (10 hours) of *S. mutans*, *S. sputigena*, and *S. sputigena-S. mutans* biofilms in addition to the 24-hour biofilms that were presented in the original submission. The data demonstrate that both *S. mutans* and *S. sputigena* can bind to the surface and grow as monocultures. Interestingly, when co-cultured, *S. sputigena* appears to form clusters in close proximity to *S. mutans* in the mixed biofilm at 10 hours (white arrowheads), suggesting early physical interactions between the two species during biofilm development. It is possible that the initial *S. sputigena* cells surrounding *S. mutans* may form the precursor of the honeycomb-like superstructure observed at 24 hours. The time course for single biofilms formed by other new species (*L. wadei* and *P. salivae*) and their mixed biofilms with *S. mutans* are also provided. We have included the additional data in Extended Data Fig. 5, which also are shown below for convenience.

Figure. A time course of biofilm development on tooth-mimetic surface. Confocal images (top views) of single- and mixed-species biofilms on saliva-coated hydroxyapatite surfaces formed by each of the new species alone and co-cultured with *S. mutans*. (A) Initial attachment stage at 4 hours. The hydroxyapatite surfaces are shown in grey. (B) Intermediate stage at 10 hours. White arrowheads, *S. sputigena* cells forming aggregates in close proximity to *S. mutans* clusters. (C) The mature biofilms at 24 hours (as shown in Fig. 5). In each panel, the upper images are single-species biofilms formed by *S. mutans* and each of the new top species, and the lower images are mixed biofilms of each new species co-cultured with *S. mutans*. Green, *S. mutans*; red, new species (*S. sputigena*, *L. wadei* or *P. salivae*) Scale bars, 20 μm .

1.11- The data suggesting *S. sputigena* becomes trapped in the glucan matrix produced by *S. mutans* are not convincing or definitive. The best data related to this point are presented in Figure 6EF. These data are only suggestive and the conclusions made by the author are speculative.

Authors' response: We thank the reviewer for raising this point. First, we must point out that the glucanohydrolases (mutanase and dextranase) used in this experiment are highly specific for hydrolyzing α -glucans produced by *S. mutans* [Ren et al., 2019]. The observation that the surface motility of *S. sputigena* was recovered following glucanohydrolase degradation suggests they were entrapped by the streptococcal α -glucans. Second, to provide additional support and further validate this observation, we conducted additional experiments as follows: We grew *S. sputigena* single-species biofilms and exogenously supplemented with purified GtfB (without bacteria), which is an *S. mutans*-derived exoenzyme that produces α -glucans [Bowen et al., 2011]. We found that *S. sputigena* cells grown in the presence of exogenous Gtf were trapped by α -glucans produced by the enzyme, showing no surface motility whereas most *S. sputigena* cells growing without Gtf were motile (Figure below). This finding together with the data in Figure 6 in the original submission supports the entrapment of *S. sputigena* by *S. mutans* Gtf derived α -glucans in the co-culture. We have included the additional data in Extended Data Fig. 6, which are shown below for convenience.

Figure. *S. sputigena* in the single-species biofilm becomes trapped in extracellular glucans produced by exogenous, cell-free GtfB enzyme. (A) Immobilized *S. sputigena* cells trapped by EPS α -glucan matrix produced by purified GtfB. Red, *S. sputigena*; cyan, α -glucan matrix. Colors indicate trajectories that originated from individual cells. Top panel, *S. sputigena* cells trapped by GtfB produced α -glucan matrix showed no mobility; bottom panel, without GtfB, *S. sputigena* cells displayed surface mobility. (F) Accumulated *S. sputigena* cell displacement (total path length) relative to the initial position. Left, accumulated displacement of *S. sputigena* cells in the presence of GtfB; right, accumulated cell displacement in the vehicle control. SS, *S. sputigena*. Scale bars, 10 μ m.

Reviewer #2:

This work addressed a global oral health problem that compromises quality of life, namely childhood dental caries. The etiological agent underlying dental caries is the microbial biofilm developing on the tooth surface, particularly when this micro-environment acquires an acidophilic and acidogenic profile. Mutans streptococci are long known to be strongly associated with the disease, yet the list of the involved novel taxa is increasing, thanks to the application of high-throughput nucleotide or amino acid sequencing techniques. This particular study used human dental biofilm samples from preschool-age children as well as an experimental rodent model. The metagenomics analysis identified 16 taxa strongly associated with childhood caries. Perhaps the main highlight of the study is the identification of the previously unrecognized ability of *S. sputigena* to exacerbate disease severity in vivo when co-infected with *S. mutans*.

The study is very well designed with multiple levels of experimentations and approaches, and is expected to have an important contribution to the literature. I would further recommend the following:

2.1- Other newly discovered important taxa have also been shown cooperate with *S. mutans*. Discuss finding of other studies whereby *Scardovia wiggisiae* in particular is shown to synergize with *S. mutans* (PMID: PMID: 28930642, PMID: 28930642)

Authors' response: We appreciate the suggestion and have now included the presentation of these recent findings regarding *Scardovia* and its interactions with *S. mutans* in the Introduction of revised manuscript.

2.2- Discuss findings of other studies whereby other commensal taxa, such as *S. oralis*, are shown to antagonize with *S. mutans* (PMID: 29327482)

Authors' response: We agree and have included additional information about *S. oralis* and its competitive interactions with *S. mutans*. In the introduction, we discussed other species, such as *S. oralis* and *Scardovia wiggisiae*, which have been shown to interact with *S. mutans*.

2.3- The title is rather generic and does not directly acknowledge the species or the disease. Therefore, “*S. sputigena*” should be added next to “pathobiont” and “oral disease” should be replaced “caries”.

Authors' response: We appreciate the reviewer’s suggestion that the title should be more attuned to oral disease under study (i.e., early childhood caries) and elevate *S. sputigena* as the pathobiont of focus. We have revised out title as follows: “***Selenomonas sputigena* as a pathobiont mediating spatial structure and biofilm virulence in early childhood caries**”

Reviewer #3:

By employing a new multimodal pipeline, in this study, the authors explored novel disease-relevant inter-species interactions and pathogenic mechanisms in preschool-age children. The authors first identified 16 taxa strongly associated with childhood caries. Among them, the authors found a novel inter-species interaction and unique spatial structuring in which a motile flagellated species (i.e., *S. sputigena*) becomes immobilized in the EPS matrix produced by disease-causing pathobionts (i.e., *S. mutans*) and proliferates to build a 3D multicellular superstructure with enhanced acidogenesis. The authors further demonstrated that the interaction between *S. sputigena* (SS) and *S. mutans* (SM) augments the severity of dental caries *in vivo*, suggesting a new pathobiont exacerbating biofilm virulence.

I enjoyed reading this manuscript. The quality of the writing and the presentation of the figures was superb and I really appreciate the authors efforts and attention to detail. Overall, I felt the overall message is clear and the work is impactful in the field, but needs to be improved with some critical experiments and logical controls to support their major conclusion. I have outlined the major and other specific concerns below:

Authors' response:

We appreciate the reviewer's evaluation of our work and the suggestions for improvement. We have conducted additional experiments to address each of them and have made revisions in the manuscript accordingly.

Major concerns:

3.1- In Figure 4, by co-culturing SS and SM, the authors showed a synergistic effect in the acid production by them. Does the SS activate the acid production from SM? or the other way around? , or both? What are the molecular mechanisms behind this metabolic activation? As the authors claimed in the text, if pathobiont-mediated spatial structuring enhances biofilm virulence, the authors should provide the data demonstrating that special structuring is required for enhancement of the virulent functions such as acid generation.

Authors' response: We appreciate the important points that are raised by the reviewer. To address these questions, we performed additional experiments and measured the *in-situ* pH at the biofilm-apatite interface, which is the critical element for disease development (Figure below). We found rapid acidification and highly acidic regions (pH<5.5) at the biofilm-apatite interface of the superstructure formed by *S. sputigena* and *S. mutans* after glucose exposure (Panel A, top). We then used both genetic (*S. mutans* *gtfBC* mutant strain) and biochemical (enzymatic) approaches to disrupt the biofilm spatial structure. Given that *S. mutans*-derived EPS α -glucans were detected throughout the entire biostructure mediating co-adhesion for the interspecies assembly (Fig. 7C and E), we used a *S. mutans* double-knockout of *gtfB* and *gtfC* (deficient in producing α -glucans [Bowen et al., 2011]) to disrupt the biofilm spatial structuring with *S. sputigena*. We found that the biofilm formed by *S. sputigena* and *S. mutans* Δ *gtfBC* mutant could not form a structured community and create acidic pH regions (Panel A, middle). This finding was further corroborated by co-culturing *S. sputigena* and *S. mutans* wild type in the presence of glucanohydrolases (dextranase and mutanase) that specifically breakdown α -glucans and dismantle the biofilm structure without affecting bacterial viability. These data demonstrate that the enzyme-treated biofilm is unable to create a localized acidic environment at the interface (Panel A, bottom). Together, the results suggest that the intact spatial structure is required to create a virulent acidic

microenvironment. We have included the additional data in Extended Data Fig. 7 in the revised Extended Data, and have copied the figure below for convenience.

Figure. Real-time pH profile at the biofilm-apatite interface. (A) *In situ* pH was assessed using a pH-responsive probe (C-SNARF-4) and high-resolution confocal imaging. Left image illustrates the biofilm structure on the surface. Images on the right show the real-time pH distribution at the interface (color-coded, pH 4.5-7.4) over time after 1% glucose exposure. White solid lines indicate the outline of the biofilm superstructure formed by *S. sputigena* and *S. mutans*. SM, *S. mutans*; SS, *S. sputigena*; PS, *P. salivae*. Scale bars, 10 μ m. (B) Quantitative pH measurements at the interface over time by ratiometric analysis. Data are plotted as mean \pm standard deviation (n=3). *, p<0.05 by one-way analysis of variance with Tukey's multiple-comparison test (t=15 min).

We agree that it is critical to understand the molecular mechanisms governing the synergism between *S. sputigena* and *S. mutans* to enhance biofilm virulence. We identified and reported glycolysis IV, an important pathway for acidogenesis, as one of the most abundant pathways that were upregulated in disease-associated biofilm transcriptomes and, notably, *S. sputigena* was implicated with upregulation of this important pathway (Extended Data Fig. 2). We believe this is an important insight that will guide our future mechanistic studies, including *in-situ* transcriptomics and metabolic interactions between *S. sputigena* and *S. mutans* within structured biofilms.

3.2- Related to the previous comment, does the Δ gtfC mutant of SM fail to enhance the biofilm virulence? How about the deletion of SS's motility-related gene flagellin in the context of the enhancement of biofilm virulence? To prove the usefulness of their experimental design shown in Figure 1, the authors should validate the functional impact of those genes (at least one related gene) *in vitro* and *in vivo*, and identify the mechanistic link. Just providing the association is not sufficient.

Authors' response: We appreciate these important points. Additional experiments were performed to test the ability of biofilms by *S. sputigena* and *S. mutans gtfBC* mutant to create acidic microenvironment, critical for biofilm virulence. We used *S. mutans Δ gtfBC* mutants because *gtfB* and *gtfC* are both critical for EPS glucan-matrix and biofilm formation, have a substantial overlap in functions and are interdependent (Bowen et al., 2011). The *in situ* pH mapping analysis showed that biofilms formed by *S. sputigena-S. mutans Δ gtfBC* mutant were unable to create an acidic niche at the biofilm-apatite interface, indicating that *gtfBC* is critical for the biofilm virulence. This observation is consistent with the inability of *S. mutans Δ gtfBC* to cause dental caries *in vivo* using a rodent model of the disease.

We agree that it is important to validate the functional impact of *S. sputigena*'s motility-related genes (especially the flagellin) on biofilm virulence. However, to the best of our knowledge, there is no *S. sputigena* mutant library readily available because this microorganism has been recalcitrant to genetic manipulation using conventional techniques. We are planning to utilize new genetic engineering methods (including the restriction modification-silent tools) to create a mutant library for *S. sputigena*, which may allow us to further understand the role of *S. sputigena* genes in polymicrobial communities.

Other concerns,

3.3- The text and content of Figure 1 do not match completely. As a result, the readers may miss where the authors describe in the manuscript. The authors should resolve this issue.

Authors' response: We thank the reviewer for pointing this out. We have revised the previous "Figure 1 top" to the current "Figure 1", as the previously "bottom" part was eliminated. Figure 1 as a whole depicts the procedure overarching our paper.

3.4- In Figure 8, to strengthen the specificity of SS, the author should provide negative controls such as *P. salivae* (PS) that do not increase the proton production rate when co-culturing with SM.

Authors' response: This is an excellent point and suggestion, and accordingly, we have performed additional experiments using an ex-vivo human enamel biofilm model that allows direct assessment of the acid-induced damage (demineralization) on the enamel surface. Given that *P. salivae* does not increase the proton production rate when co-cultured with *S. mutans* (Fig. 4, A2), we used *P. salivae-S. mutans* co-culture as a control, as suggested by the reviewer. We compared the enamel damage caused by *S. mutans-S. sputigena* co-culture vs. *S. mutans* alone (Figure below). Macroscopically, we found large areas of enamel demineralization associated with *S. mutans-S. sputigena* biofilm, characterized by chalky-like opaque surface under the stereoscope (Panel A, bottom left), similar to those found in early childhood caries lesions. In contrast, only small areas of opaque demineralized areas were found on the enamel surface from *P. salivae-S. mutans* (Panel A, middle left), and *S. mutans* alone (Panel A; upper left) biofilms. These differences were confirmed using confocal topography imaging and transverse microradiography analysis. The enamel surfaces from *S. mutans-S. sputigena* biofilms showed higher roughness compared to those from *S. mutans* biofilms and *P. salivae-S. mutans* biofilms (Panel C,

“Roughness of enamel surface”; $P < 0.05$), with widespread regions of surface damage (Panel A, lower right) . In contrast, the surface roughness from *P. salivae*-*S. mutans* biofilms was similar to that from *S. mutans* alone (Panel C; $P > 0.05$), both showing milder enamel surface demineralization (Panel A, right).

Transverse microradiography analysis validated the extent of enamel acid-induced damage (i.e., decay or caries lesion), showing significantly higher mineral loss and deeper lesions caused by *S. mutans*-*S. sputigena* biofilms ($P < 0.05$) versus *P. salivae*-*S. mutans* or *S. mutans* biofilms (Panels B and C). Together, these data suggest specificity of the *S. sputigena*-*S. mutans* interaction and spatial structuring to promote biofilm virulence. We have included the additional data in Extended Data Fig. 8 in the revised Extended Data, which are copied below for convenience.

	Roughness of enamel surface (μm)	Mineral loss ($\% \times \mu\text{m}$)	Lesion depth (μm)
SM	0.09 ± 0.03^a	2680.0 ± 553.0^a	73.7 ± 16.1^a
SM+PS	0.18 ± 0.14^a	2335.0 ± 507.7^a	70.5 ± 5.0^a
Sm+Ss	0.51 ± 0.16^b	3757.5 ± 145.7^b	101.8 ± 5.7^b

Figure. *S. sputigena*-*S. mutans* biofilm causes tooth-decay on human enamel. (A) Multi-scale analyses of the human tooth-enamel underneath the biofilms. Left, macroscopic demineralized, white spot-like lesions (brighter, chalky areas) developed on the enamel surface inflicted by *S. mutans* alone (SM), *S. mutans*-*P. salivae* co-culture (SM-PS), and *S. mutans*-*S. sputigena* co-culture (SM-SS); right, corresponding surface topography analysis showing micro cavities formed on the enamel surface. The enamel surface topography is color-coded to visualize the micro cavities. (B) Transverse microradiography of the human enamel surface underneath the biofilms. The average lesion depth of the enamel demineralization caused by biofilms is indicated in the image. (C) Quantitative analysis of tooth surface topography (roughness) and tooth mineral analysis (mineral loss and lesion depth). The numbers are mean \pm standard deviation (n=4). Groups that do not share the same lowercase letter are significantly different ($p < 0.05$ by one-way analysis of variance with Tukey's multiple-comparison test)

REVIEWERS' COMMENTS

Reviewer #2 (Remarks to the Author):

The authors have sufficiently addressed the reviewer's points.

Reviewer #3 (Remarks to the Author):

The authors have made substantial improvements in the paper. I was impressed with the quality of the results from the measurement of the in-situ pH at the biofilm-apatite interfaces and subsequent mutant experiments, which strongly supports the central claims of this paper.

Reviewer #4 (Remarks to the Author):

The authors have adequately addressed this reviewer's concerns and the manuscript was considerably improved. Therefore, I consider this manuscript suitable for publication in its current version.